# LINES OF THOUGHT IN LARGE LANGUAGE MODELS

**Raphaël Sarfati**
School of Civil and Environmental Engineering
Cornell University, USA
raphael.sarfati@cornell.edu

**Toni J.B. Liu**
Department of Physics
Cornell University, USA
toni.liu@cornell.edu

**Nicolas Boullé**
Department of Mathematics
Imperial College London, UK
n.boulle@imperial.ac.uk

**Christopher J. Earls**
Center for Applied Mathematics
School of Civil and Environmental Engineering
Cornell University, USA
earls@cornell.edu

## ABSTRACT

Large Language Models achieve next-token prediction by transporting a vectorized piece of text (prompt) across an accompanying embedding space under the action of successive transformer layers. The resulting high-dimensional trajectories realize different contextualization, or 'thinking', steps, and fully determine the output probability distribution. We aim to characterize the statistical properties of ensembles of these '*lines of thought*.' We observe that independent trajectories cluster along a low-dimensional, non-Euclidean manifold, and that their path can be well approximated by a stochastic equation with few parameters extracted from data. We find it remarkable that the vast complexity of such large models can be reduced to a much simpler form, and we reflect on implications. Code for trajectory generation, visualization, and analysis is available on Github at https://github.com/rapsar/lines-of-thought.

## 1 INTRODUCTION

How does a large language model (LLM) think? In other words, how does it abstract the prompt *"Once upon a time, a facetious"* to suggest adding, e.g., *"chatbot"*, and, by repeating the operation, continue on to generate a respectable fairy tale *à la* Perrault? What we know is by design. A piece of text is mapped into a set of high-dimensional vectors, which are then transported across their embedding (latent) space through successive transformer layers (Vaswani et al., 2017), each allegedly distilling different syntactic, semantic, informational, contextual aspects of the input (Valeriani et al., 2023; Song & Zhong, 2024). The final position is then projected onto an embedded vocabulary to create a probability distribution about what the next word should be. Why these vectors land where they do eludes human comprehension due to the concomitant astronomical numbers of arithmetic operations which, taken individually, do nothing, but collectively confer the emergent ability of language.

Our inability to understand the inner workings of LLMs is problematic and, perhaps, worrisome. While LLMs are useful to write college essays or assist with filing tax returns, they are also often capricious, disobedient, and hallucinatory (Sharma et al., 2023; Zhang et al., 2023). That's because, unlike traditional 'if-then' algorithms, instructions have been only loosely, abstractly, encoded in the structure of the LLM through machine learning, that is, without human intervention.

In return, language models, trained primarily on textual data to generate language, have demonstrated curious abilities in many other domains (in-context learning), such as extrapolating time series (Gruver et al., 2024; Liu et al., 2024), writing music (Zhou et al., 2024), or playing chess (Ruoss et al., 2024). Such emergent, but unpredicted, capabilities lead to questions about what other abilities LLMs may possess. For these reasons, current research is attempting to break down internal processes to make LLMs more *interpretable*.[1] Recent studies have notably revealed some aspects of

---

[1]And, eventually, more reliable and predictable.

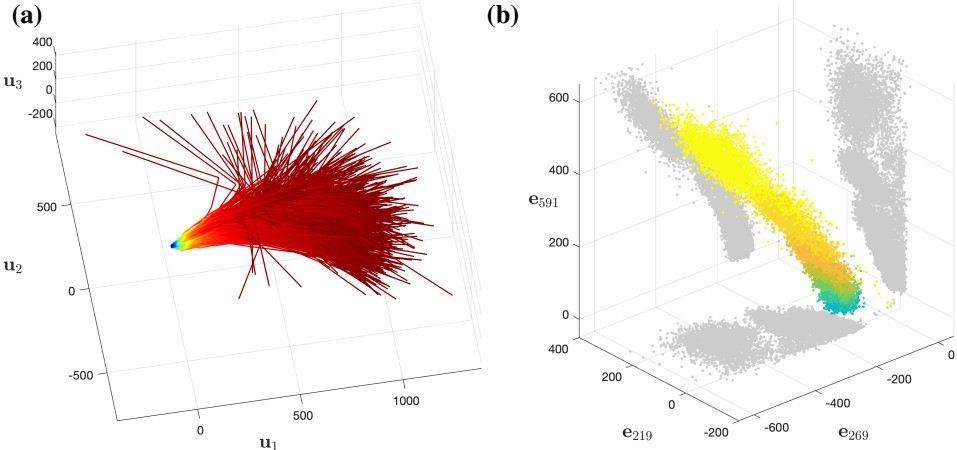

Figure 1: **(a)** Lines of thought (blue to red) for an ensemble of 1000 pseudo-sentences of 50 tokens each, projected along the first 3 singular vectors after the last layer ($t = 24$). They appear to form a tight bundle, with limited variability around a common average path. **(b)** Representation of the low-dimensional, ribbon-shaped manifold in $\mathcal{S}$ (projected along 3 Cartesian coordinates). Positions are plotted for $t = 12$ (green) to $t = 24$ (yellow).

the self-attention mechanism (Vig, 2019), patterns of neuron activation (Bricken et al., 2023; Templeton et al., 2024), signatures of 'world models'[2] (Gurnee & Tegmark, 2023; Marks & Tegmark, 2023), geometrical relationships between concepts (Jiang et al., 2024), or proposed mathematical models of transformers (Geshkovski et al., 2024).

This work introduces an alternative approach inspired by physics, treating an LLM as a complex dynamical system. We investigate which large-scale, ensemble properties can be inferred experimentally without concern for the 'microscopic' details.[3] Specifically, we are interested in the trajectories, or 'lines of thought' (LoT), that embedded tokens realize in the latent space when passing through successive transformer layers (Aubry et al., 2024). By splitting a large input text into $N$-token sequences, we study LoT *ensemble* properties to shed light on the internal, average processes that characterize transformer transport.

We find that, even though transformer layers perform $10^6 - 10^9$ individual computations, the resulting trajectories can be described with far fewer parameters. In particular, we first identify a low-dimensional manifold that explains most of LoT transport (see Fig. 1). Then, we demonstrate that trajectories can be well approximated by an average linear transformation, whose parameters are extracted from ensemble properties, along with a random component with well characterized statistics. Eventually, this allows us to describe trajectories as a kind of diffusive process, with a linear drift and a modified stochastic component.

**Main contributions.**

1. We provide a framework to discover low-dimensional structures in an LLM's latent space.

2. We find that token trajectories cluster on a non-Euclidean, low-dimensional manifold.

3. We introduce a stochastic model to describe trajectory ensembles with few parameters and extend them to continuous paths.

---

[2]World models refers to evidence of (abstract) internal representations which allow LLMs an apparent understanding of patterns, relationships, and other complex concepts.

[3]Such as: semantic or syntactic relationships, architecture specificities, etc.

## 2   METHODS

This section describes our algorithm for generating and analyzing an ensemble of tokens trajectories in the latent space of LLMs. Our code is provided in the corresponding Github repository (Sarfati et al.).

**Language models.**   We rely primarily on the 355M-parameter ('medium') version of the GPT-2 model (Radford et al., 2019). It presents the core architecture of ancestral (circa 2019) LLMs: transformer-based, decoder-only.[4] It consists of $N_L = 24$ transformer layers[5] operating in a latent space $\mathcal{S}$ of dimension $D = 1024$. The vocabulary $\mathcal{V}$ contains $N_{\mathcal{V}} = 50257$ tokens. A layer normalization (Ba et al., 2016) is applied to the last latent space position before projection onto $\mathcal{V}$ to form the logits. (This final normalization is not included in our trajectories.) We later extend our analysis to the Llama 2 7B (Touvron et al., 2023), Mistral 7B v0.1 (Jiang et al., 2023), and small Llama 3.2 models (1B and 3B) (MetaAI, 2024).

**Input ensembles.**   We study statistical properties of trajectory ensembles obtained by passing a set of input prompts through GPT-2. We generate inputs by tokenizing (Wolf et al., 2020) a large text and then chopping it into 'pseudo-sentences', i.e., chunks of a fixed number of tokens $N_k$ (see Algorithm 1). Unless otherwise noted, $N_k = 50$. These *non-overlapping* chunks are consistent in terms of token cardinality, and possess the structure of language, but have various meanings and endings (see Appendix A.1). The main corpus in this study comes from Henry David Thoreau's *Walden*, obtained from the Gutenberg Project (Project Gutenberg, 2024).[6] We typically use a set of $N_s \simeq 3000\text{--}14000$ pseudo-sentences.

**Trajectory collection.**   We form trajectories by collecting the successive vector outputs, within the latent space, after each transformer layer (`hidden_states`). For conciseness, we identify layer number with a notional 'time', $t$. Even though all embedded tokens of a prompt voyage across the latent space, only the embedding corresponding to the last token form the logits (by projection onto $\mathcal{V}$) for next-token inference. Hence, here, we only consider the trajectory of this last (or 'pilot') token. The trajectory $\boldsymbol{M}_k$ of sentence $k$'s pilot is the sequence of $24$ successive time positions $\{\boldsymbol{x}_k(1), \boldsymbol{x}_k(2), \ldots, \boldsymbol{x}_k(24)\}$, concatenated as a column matrix (Algorithm 1).

**Latent space bases.**   The latent space is spanned by the Cartesian basis $\mathcal{E} = \{\boldsymbol{e}_i\}_{i=1\ldots D}$ (the orthogonal set of one-hot unit vectors with a 1 in $i^{\text{th}}$ position, 0 elsewhere). Additionally, we will often refer to the bases $\mathcal{U}(t) = \{\boldsymbol{u}_i^{(t)}\}_{i=1\ldots D}$ formed by the left-singular vectors of the singular value decomposition (SVD) of the $D \times N_s$ matrix after layer $t$: $\boldsymbol{M} = \boldsymbol{U}\boldsymbol{\Sigma}\boldsymbol{V}^\top$, with $\boldsymbol{M}_{:,k}^{(t)} = \boldsymbol{x}_k(t)$. Vectors $\boldsymbol{u}_i$ are organized according to their corresponding singular values, $\sigma_i$, in descending order. Note that because trajectory clusters evolve over time there are 24 distinct bases.

## 3   RESULTS

We present and characterize results pertaining to ensembles of trajectories as they travel within the latent space $\mathcal{S}$.

### 3.1   LINES OF THOUGHT CLUSTER ALONG SIMILAR PATHWAYS

We endeavor to visualize and characterize the trajectories of pilot tokens in the latent space $\mathcal{S}$. The high dimensionality makes it non-trivial. Which projections should we consider?

---

[4]Compared to current state-of-the-art models, GPT-2 medium is rather unsophisticated. Nevertheless, it works. It produces cogent text that addresses the input prompt. Hence, we consider the model already contains the essence of modern LLMs and leverage its agility and transparency for scientific insight.

[5](LayerNorm +) Self-attention then (LayerNorm +) Feed-forward, with skip connections around both.

[6]The idea of using a literary piece to probe statistics of language was investigated by Markov back in 1913 (Markov, 2006).

---

**Algorithm 1** Trajectory generation in transformer-based model

---

1: **Input:** Large text: "It was the best of times, it was the worst of times, it was the age . . . "
2: Tokenize text into token sequence: $[1027, 374, 263, 1267, 287, 1662, 12, . . .]$
3: Split token sequence into $n$-token pseudo-sentences:

$$s_1 = [1027, 374, 263], \quad s_2 = [1267, 287, 1662], \quad . . .$$

4: **for** each pseudo-sentence $s_i$ **do**
5:     Semantic embedding:

$$\boldsymbol{E}_S = [\boldsymbol{v}(1027), \boldsymbol{v}(374), \boldsymbol{v}(263)] \quad \text{for } s_1$$

6:     $\boldsymbol{E}^0 = \boldsymbol{E}_S + \boldsymbol{E}_P$ {add positional embeddings $\boldsymbol{P}$}
7:     **for** $t = 1 \rightarrow 23$ **do**
8:         $\boldsymbol{E}^{t+1} = \text{TransformerLayer}_t(\boldsymbol{E}^t)$ {update embeddings through transformer layer}
9:         $\boldsymbol{x}(t+1) = \boldsymbol{E}^{(t+1)}_{:,\text{end}}$ {extract last token representation}
10:         $\boldsymbol{M}_{:,t+1} = \boldsymbol{x}(t+1)$ {save trajectory array}
11:     **end for**
12: **end for**
13: **Output:** Final embeddings $\boldsymbol{x}(t+1)$ for all pseudo-sentences

---

There is no reason a priori for the Cartesian axes, $\boldsymbol{e}_i$, to align with any meaningful directions, so we seek relevant alternative bases informed by the data. Naturally, we consider the bases $\mathcal{U}(t)$ formed by the singular vectors of pilots ensemble after each layer $t$.

Using these bases aligned with the data's intrinsic directions, we observe in Fig. 1a that trajectories tend to cluster together, instead of producing an isotropic and homogeneous filling of $\mathcal{S}$. Indeed, LoTs for different, *independent* pseudo-sentences follow a common path (forming *bundles*), augmented by individual variability. Specifically, there exist directions with significant displacement relative to the spread (mean over standard deviation); and positions at different times form distinct clusters, as shown in Fig. A1.

## 3.2 LINES OF THOUGHT FOLLOW A LOW-DIMENSIONAL MANIFOLD

We remark in Fig. 2a that the intrinsic bases $\mathcal{U}(t)$ rotate only slightly across successive timepoints $t$. Besides, Fig. 2b shows that the corresponding singular values decay quickly over several orders of magnitude. Both suggest that LoTs may be described by a low-dimensional curved subspace.

But how many dimensions are relevant? Singular values relate to ensemble variance along their corresponding directions. Since the embedding space is high-dimensional, however, the curse of dimensionality looms, hence the significance of Euclidean distances crumbles. To circumvent this limitation, we consider a more practical metric: how close to the original output distribution on the vocabulary does a reduction in dimensionality get us?

To investigate this question, we express token positions $\boldsymbol{x}(t)$ in the singular vector basis $\mathcal{U}(t)$:

$$\boldsymbol{x}(t) = \sum_{i=1}^{K} a_i^{(t)} \boldsymbol{u}_i^{(t)},$$

where the $\boldsymbol{u}_i^{(t)}$'s are organized by descending order of their corresponding singular values. By default $K = D$, and the true output distribution $\boldsymbol{p}^{\mathcal{V}}$ is obtained. Now, we examine what happens when, instead of passing the full basis set, we truncate it, *after each layer*, to keep only the first $K < D$ principal components. We compare the resulting output distribution, $\mathbf{p}_K^{\mathcal{V}}$ to the true distribution $\mathbf{p}^{\mathcal{V}}$ using KL divergence $D_{\text{KL}}(\mathbf{p}_K^{\mathcal{V}} \| \mathbf{p}^{\mathcal{V}})$. In Fig. 2c, we see that $D_{\text{KL}}$ decreases very slowly with decreasing $K$, up to about $K_0 = 256$. At that point, $D_{\text{KL}}$ is only about 10% of its uncorrelated baseline value, implying that most of the true distribution is recovered when keeping only about $K_0 = 256$, or 25%, of the principal components. In other words, for the purpose of next-token prediction, LoTs are quasi-256-dimensional.

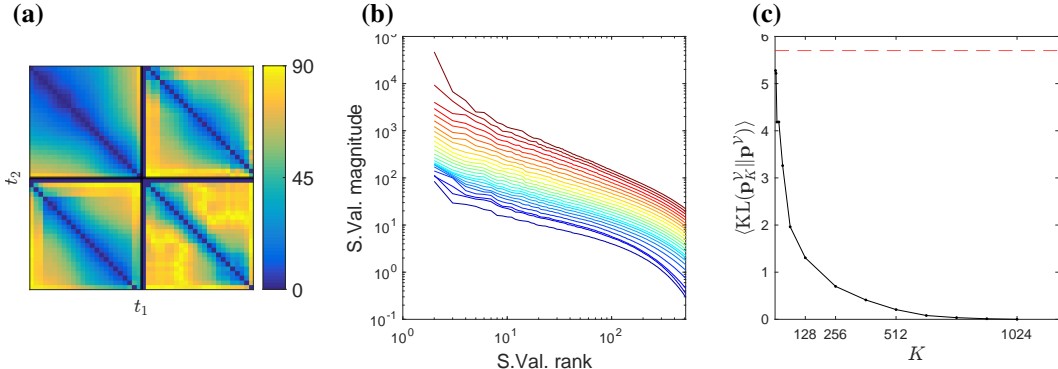

Figure 2: **(a)** Angle between the first 4 singular vectors at $(t_1, t_2)$, $\arccos(\boldsymbol{u}_i^{(t_1)} \cdot \boldsymbol{u}_i^{(t_2)})$, for $i = \{1, 2, 3, 4\}$ (top-left, top-right, bottom-left, bottom-right, respectively). **(b)** Singular values for $t = 1, \dots, 24$ (blue to red). Clusters stretch more and more after each layer. The leading singular values, $\sigma_1(t)$, have been omitted for clarity. **(c)** Average (over all trajectories) KL divergence between reduced dimensionality trajectories output and true output distributions, as the dimensionality $K$ is increased. The red dashes line shows the average KL divergence for output distributions from unrelated inputs
(baseline for dissimilar distributions).

If these principal directions remained constant at each layer, this would imply that $75\%$ of the latent space could be discarded with no consequence. This seems unrealistic. In fact, the principal directions rotate slightly over time, as displayed in Fig. 2a. Eventually, between $t = 1$ and $t = 24$, the full Cartesian basis $\mathcal{E}$ is necessary to express the first singular directions. Thus, we conclude that **lines of thoughts evolve on a low-dimensional curved manifold** of about 256 dimensions, that is contained within the full latent space (Fig. 1b).

### 3.3 LINEAR APPROXIMATION OF TRAJECTORIES

Examination of the singular vectors and values at each time step indicates that LoT bundles rotate and stretch smoothly after passing through each layer (Fig. 2). This suggests that token trajectories could be *approximated* by the linear transformations described by the ensemble, and extrapolated accordingly, from an initial time $t$ to a later time $t + \tau$. Evidently, it is improbable that a transformer layer could be replaced by a mere linear transformation. We rather hypothesize that, in addition to this deterministic average path, a token's location after layer $t + \tau$ will depart from its linear approximation from $t$ by an unknown component $\boldsymbol{w}(t, \tau)$.[7] We propose the following model:

$$\boldsymbol{x}(t + \tau) = \boldsymbol{R}(t + \tau)\boldsymbol{\Lambda}(t, \tau)\boldsymbol{R}(t)^\top \boldsymbol{x}(t) + \boldsymbol{w}(t, \tau), \qquad (1)$$

where $\boldsymbol{x}(t)$ is the pilot token's position *in the Cartesian basis*, and $\boldsymbol{R}, \boldsymbol{\Lambda}$ are rotation (orthonormal) and stretch (diagonal) matrices, respectively. Eq. (1) formalizes the idea that, to approximate $\boldsymbol{x}(t + \tau)$, given $\boldsymbol{x}(t)$, we first project $\boldsymbol{x}$ in the ensemble intrinsic basis at $t$ ($\boldsymbol{R}^\top \boldsymbol{x}$), then stretch the coordinates by the amount given by $\boldsymbol{\Lambda}$, and finally rotate according to how much the singular directions have rotated between $t$ and $t + \tau$, $\boldsymbol{R}(t + \tau)$ (see also Fig. A2 in Appendix B). Consequently, we can express these matrices as a function of the set of singular vectors ($\boldsymbol{U}$) and values ($\boldsymbol{\Sigma}$):

$$\boldsymbol{R}(t) = \boldsymbol{U}(t), \quad \boldsymbol{\Lambda}(t, \tau) = \text{diag}(\sigma_i(t + \tau)/\sigma_i(t)) = \boldsymbol{\Sigma}(t + \tau)\boldsymbol{\Sigma}^{-1}(t).$$

Fig. 3 shows the close agreement, at the ensemble level, between the true and extrapolated positions.

This is confirmed by the observation that the two sets are not separable with a trained linear classifier, including at large $\tau$ (see Appendix). Eq. (1) is merely a linear approximation as it is similar to assuming that LoT clusters deform like an elastic solid, where each point maintains the same vicinity, as illustrated in Fig. A2. The actual coordinates ought to include an additional random component $\mathbf{w}(t, \tau)$, which *a priori* depends on both $t$ and $\tau$.

---

[7]We emphasize that prompt trajectories are completely deterministic; the stochastic component introduced in the model accounts for the fact that we perform a linear extrapolation based only on a token's position at a certain time, which unsurprisingly deviates from the true position obtained from processing the full prompt with transformer layers.

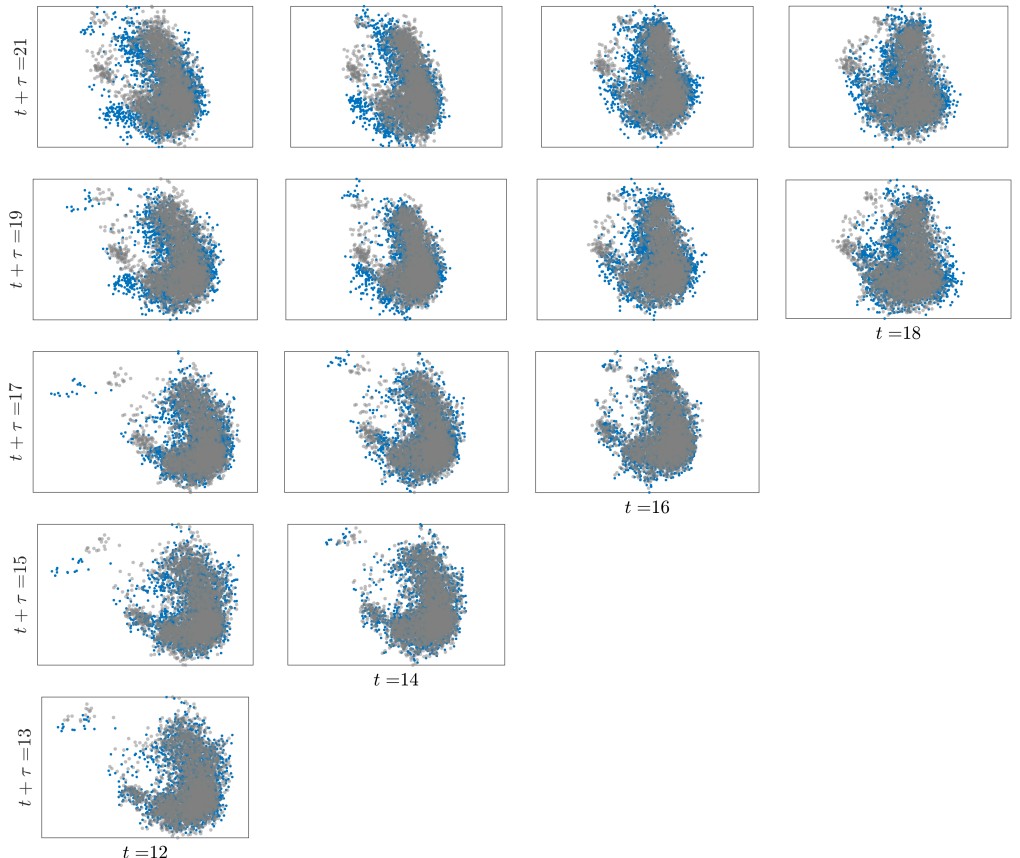

Figure 3: Extrapolated token positions $\tilde{\boldsymbol{x}}^{(k)}$ (blue) from $t = \{12, 14, 16, 18\}$ to $t + \tau = \{t + 1, \ldots, 21\}$, compared to their true positions $\boldsymbol{x}^{(k)}$ (gray), projected in the $(\boldsymbol{u}_2^{(t)}, \boldsymbol{u}_3^{(t)})$ planes.

Is it possible to express $\boldsymbol{w}$ in probabilistic terms? We consider the empirical residuals

$$\delta\boldsymbol{x}(t, \tau) = \boldsymbol{x}(t + \tau) - \tilde{\boldsymbol{x}}(t, \tau)$$

between true positions $\boldsymbol{x}$ and linear approximations $\tilde{\boldsymbol{x}}(t, \tau) = \boldsymbol{R}(t + \tau)\boldsymbol{\Lambda}(t, \tau)\boldsymbol{R}(t)^\top\boldsymbol{x}(t)$. We investigate the distributions and correlations of $\delta\boldsymbol{x}(t, \tau)$ across layer combinations $(t, t + \tau)$.

From the data, Fig. 4 shows that, for all $(t, t + \tau) \in \{1, \ldots, 23\} \times \{t + 1, \ldots, 24\}$, the ensemble of $\delta\boldsymbol{x}(t, \tau)$ has the following characteristics: 1) it is Gaussian, 2) with zero mean, 3) and variance scaling as $\exp(t + \tau)$. In addition, Fig. A3 shows that the distribution is isotropic, with no evidence of spatial cross-correlations . Hence, we propose:

$$\mathrm{w}_i(t, \tau) \sim \mathcal{N}(0, \alpha e^{\lambda(t+\tau)}), \tag{2}$$

i.e., each coordinate $\mathrm{w}_i$ of $\boldsymbol{w}$ is a Gaussian random variable with mean zero and variance $\alpha e^{\lambda(t+\tau)}$. Linear fitting of the logarithm of the variance yields $\alpha \simeq 0.64$ and $\lambda \simeq 0.18$. Even though this formulation ignores some variability across times and dimensions, it is a useful minimal modelling form to describe the ensemble dynamics with as few parameters as possible.

### 3.4 LANGEVIN DYNAMICS FOR CONTINUOUS TIME TRAJECTORIES

Just like the true positions $\boldsymbol{x}(t)$, matrices $\boldsymbol{R}$ and $\boldsymbol{\Lambda}$ are known (empirically) only for integers values of $t$.[8] Can we extend Eq. (1) to a continuous time parameter $t \in [1, 24]$? Indeed, it is possible to *interpolate* $\boldsymbol{R}$ and $\boldsymbol{\Lambda}$ between their known values (Absil et al., 2008). Specifically, $\boldsymbol{R}(t)$ remains orthogonal and rotates from its endpoints; singular values can be interpolated by a spline function.

---

[8]That is, after each layer.

In return, this allows us to interpolate trajectories between transformer layers.[9] Thus, we extend Eq. (1) to a continuous time variable $t$, and write in infinitesimal terms the Langevin equation for the dynamics:

$$d\boldsymbol{x}(t) = \left[ \dot{\boldsymbol{R}}(t)\boldsymbol{R}(t)^\top + \boldsymbol{R}(t)\dot{\boldsymbol{S}}(t)\boldsymbol{R}(t)^\top \right] \boldsymbol{x}(t)\, dt + \sqrt{\alpha\lambda\exp(\lambda t)}\, d\mathbf{w}(t), \tag{3}$$

where $\dot{\boldsymbol{S}} = \mathrm{diag}\left(\dot{\sigma}_i/\sigma_i\right)$ and $d\mathbf{w}(t)$ is a differential of a Wiener process (Pavliotis, 2014). We defer the mathematical derivation to Appendix A.2. This equation artificially extends LoTs to continuous paths across $\mathcal{S}$. It provides a stochastic approximation to any token's trajectory, at all times $t$.

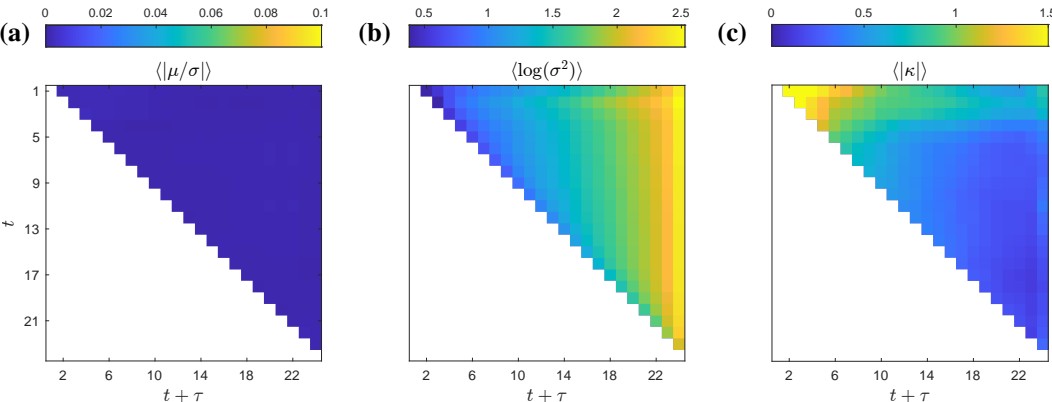

Figure 4: Statistics of $\delta\boldsymbol{x}(t,\tau)$: mean $\mu$, variance $\sigma^2$, excess kurtosis $\kappa$. Brackets $\langle\dots\rangle$ denote average over directions $\boldsymbol{e}_i$ (see Fig. A4 for details). **(a)** For all $(t, t+\tau)$, $\mu \simeq 0$ (that is, $\mu/\sigma \ll 1$). **(b)** $\log(\sigma^2)$ increases linearly in time, only depends on $t+\tau$. **(c)** The *excess* kurtosis (kurtosis minus 3) remains close to 0, indicating Gaussianity (except in early layers).

### 3.5 FOKKER-PLANCK FORMULATION

Eq. (3) is a stochastic differential equation (SDE) describing individual trajectories with a random component. Since the noise distribution is well characterized (see Eq. (2)), we can write an equivalent formulation for the *deterministic* evolution of the probability density $P(\boldsymbol{x}, t)$ of tokens $\boldsymbol{x}$ over time (Pavliotis, 2014). The Fokker-Planck equation[10] associated to Eq. (3) reads:

$$\frac{\partial P(\boldsymbol{x}, t)}{\partial t} = -\nabla_{\boldsymbol{x}} \cdot \left[ \left( \dot{\boldsymbol{R}}\boldsymbol{R}^\top + \boldsymbol{R}\dot{\boldsymbol{S}}\boldsymbol{R}^\top \right) \boldsymbol{x} P(\boldsymbol{x}, t) \right] + \frac{1}{2}\alpha\lambda e^{\lambda t}\, \nabla_{\boldsymbol{x}}^2 P(\boldsymbol{x}, t). \tag{4}$$

This equation captures trajectory ensemble dynamics in a much simpler form, and with far fewer parameters, than the computation actually performed by the transformer stack on the fully embedded prompt. The price paid for this simplification is a probabilistic, rather than deterministic, path for LoTs. We now test our model and assess the extent and limitations of our results.

## 4 TESTING AND VALIDATION

### 4.1 SIMULATIONS OF THE STOCHASTIC MODEL

We test our continuous-time model described above. Due to the high dimensionality of the space, numerical integration of the Fokker-Planck equation, Eq. (4), is computationally prohibitive. Instead, we simulate an ensemble of trajectories based on the Langevin formulation, Eq. (3). The technical details are provided in Appendix A.3.

The results presented in Fig. 5 show that the simulated ensembles closely reproduce the ground truth of true trajectory distributions. We must note that Eqs. (3) and (4) are not path-independent;

---

[9]These interpolated positions do not hold any interpretive value, but may be insightful for mathematical purposes.

[10]Also known as Kolmogorov forward equation.

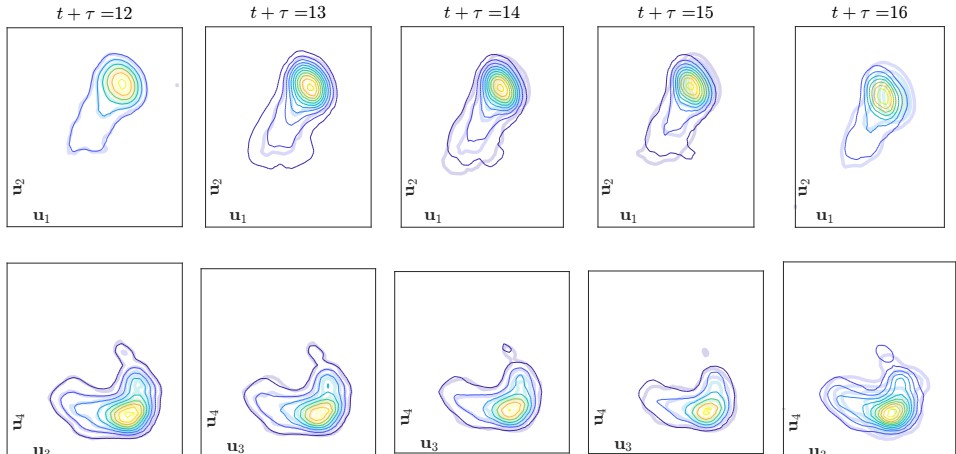

Figure 5: Simulated distributions for $t = 12$, $t + \tau = \{12, 13, 14, 15, 16\}$, projected on the $(\boldsymbol{u}_1, \boldsymbol{u}_2)$ plane (top row) and the $(\boldsymbol{u}_3, \boldsymbol{u}_4)$ plane (bottom row). Distributions have been approximated from ensemble trajectories, 10 trajectories for each initial point. Background lines indicate true distributions, thin lines on top indicate simulations.

therefore, their solution depend on the value of $\boldsymbol{R}(t)$, $\boldsymbol{S}(t)$ at all time $t$. Since there is no 'true' value for the matrices in-between layers, the output of numerical integration naturally depends on the interpolation scheme. Hence, discrepancies are to be expected.

## 4.2 NULL TESTING

We now examine trajectory patterns for **non-language inputs** and **untrained models**.

### 4.2.1 GIBBERISH

We generate non-language ('gibberish') pseudo-sentences by assembling $N$-token sequences of random tokens in the vocabulary, and pass them as input to GPT-2. The resulting trajectories also cluster around a path similar to that of language. However, the two ensembles, language and gibberish, are linearly separable at all layers (see Fig. A5 in Appendix B.6), indicating that they travel on two distinct, yet adjacent, manifolds.

### 4.2.2 UNTRAINED & ABLATED MODELS

We compare previous observations with the null baseline of an untrained model.

First, we collect trajectories of the *Walden* ensemble passing through a reinitialized version of GPT-2 (the weights have been reset to a random seed). We observe that while LoTs get transported away from their starting point, the trajectories follow straight, quasi-parallel paths, maintaining their vicinity (see Fig. A5). Furthermore, the model of Eqs. (1) and (2) does not hold; Fig. A6 shows that the variance of $\delta\boldsymbol{x}$ does not follow the $\exp(t + \tau)$ scaling, and the distributions are far from Gaussian.

Next, we consider an ablated model, where only layers 13 to 24 have been reinitialized. When reaching the untrained layers, the trajectories stop and merely diffuse about their $t = 12$ location (Fig. A5).

In conclusion, upon training, the weights evolve to constitute a specific type of transport in the latent space.

## 4.3 RESULTS WITH OTHER MODELS

We repeat the same approach with a set of larger and more recent LLMs. We collect the trajectories of the *Walden* ensemble in their respective latent spaces.

**Llama 2 7B.** We first investigate the Llama 2 7B model (Touvron et al., 2023).[11] Remarkably, the pattern of GPT-2 repeats. Token positions at $t + \tau$ can be extrapolated from $t$ by rotation and stretch using the singular vectors and values of the ensemble. The residuals are distributed as those of GPT-2, with $\mathrm{w}_i(t, \tau) \sim \mathcal{N}(0, \alpha e^{\lambda(t+\tau)})$, see Fig. A7. The values for the parameters $\alpha$ and $\lambda$, however, differ from those of GPT-2 (here, $\alpha \simeq -5.4, \lambda \simeq 0.27$).

**Mistral 7B.** Trajectories across the Mistral 7B (v0.1) model (Jiang et al., 2023)[12] also follow the same pattern (Fig. A8). We note, however, that Eq. (4) only holds up until layer 31. It seems as though the last layer is misaligned with the rest of the trajectories, as linear extrapolation produces an error that is much larger than expected.

**Llama 3.2.** The last layer anomaly is also apparent for Llama 3.2 1B[13], both in the mean and variance of $\delta \boldsymbol{x}(t, 16)$ (see Fig. A9). However, the rest of the trajectories follows Eq. (1). The same pattern is observed for Llama 3.2 3B[14] in Fig. A10.

It is noteworthy that these three recent models feature the same anomaly at the last layer. The reason is not immediately evident, and perhaps worth investigating further. In addition, we remark that all models also show deviations from predicted statistics across the very first layers (top-left corners). We conjecture that these anomalies might be an effect of re-alignment or fine-tuning, as the first and last layers are the most exposed to perturbations which might not propagate deep into the stack.

## 5 CONCLUSION

**Summary.** This work began with the prospect of visualizing token trajectories in their embedding space $\mathcal{S}$. The space is not only high-dimensional, but also isotropic: all coordinates are *a priori* equivalent.[15] Hence, we sought directions and subspaces of particular significance in shaping token trajectories[16], some kind of 'eigenvectors' of the transformer stack.

Instead of spreading chaotically, lines of thought travel along a low-dimensional manifold. We used this pathway to extrapolate token trajectories from a known position at $t$ to a later time, based on the geometry of the ensemble. Individual trajectories deviate from this average path by a random amount *with well-defined statistics*. Consequently, we could interpolate token dynamics to a continuous time in the form of a stochastic differential equation, Eq. (3). The same ensemble behavior holds for various transformer-based pre-trained LLMs, but collapses for untrained (reinitialized) ones.

This approach aims to extract important features of language model internal computation. Unlike much of prior research on interpretability, it is agnostic to the syntactic and semantic aspects of inputs and outputs. We also proposed geometrical interpretations of ensemble properties which avoid relying on euclidean metrics, as they become meaningless in high-dimensional spaces.

**Limitations.** This method is limited to open-source models, as it requires extracting hidden states; fine-tuned, heavily re-aligned models might exhibit different patterns. In addition, it would be compelling to connect the latent space with the space of output distributions, for example by investigating the relative arrangement of final positions with respect to embedded vocabulary. However, this is complicated by the last layer normalization which typically precedes projection onto the vocabulary. This normalization has computational benefits, but its mathematical handling is cumbersome: it is highly non-linear as it involves the mean and standard deviation of the input vector.

---

[11]Decoder-only, 32 layers, 4096 dimensions; released July 2023 by Meta AI.

[12]Decoder-only, 32 layers, 4096 dimensions; released September 2023 by Mistral AI.

[13]Decoder-only, 16 layers, 2048 dimensions; released September 2024 by Meta AI.

[14]Decoder-only, 28 layers, 3072 dimensions; released September 2024 by Meta AI.

[15]Unlike other types of datasets where different dimensions might have well-defined meaning, for example: temperature, pressure, wind speed, etc.

[16]And hence defining next-token distribution outputs

**Implications.** Just like molecules in a gas or birds in a flock, the complex system formed by billions of artificial neurons in interaction exhibits some simple, macroscopic properties. It can be described by ensemble statistics with a well defined random component. Previously, Aubry et al. (2024) had also uncovered specific dynamical features, notably *token alignment*, in transformer stacks of a wide variety of trained models.

Patterns are explanatory. Our concern here has been primarily to discover some of the mechanisms implicitly encoded in the weights of trained language models. Yet, there are also concrete and potentially practical implications to our findings.

For interpretability, finding low-dimensional structures is consequential. It is one of the most efficient ways to break down the inherent complexity of large models into more elementary constituents. Our dynamical system approach reveals a surprising dimensionality reduction of token embeddings. It suggests, notably, that the true "meaning" of embeddings is contained within individual variability (possibly orthogonal to the average pathway collectively followed by all LoTs). This is also merely a first-order approximation, which could be extended to more complete and precise equations, where the "noise term" becomes smaller and smaller. Eventually, we anticipate the possibility for hybrid architectures where the deterministic part of trajectories is delegated to a small system of equations, while the variable part, where meaning is encoded, is handled by a neural network; potentially with many fewer weights.

Our theoretical model in Eqs. (3) and (4) not only reveals low-dimensionality, but also extends token trajectories to continuous paths. In the past, the Neural Ordinary Differential Equation paradigm by Chen et al. (2019) showed that converting a discrete neural network into a continuous dynamical system had many advantages. Notably, it offers opportunities for compression and stability, while pointing towards efficient hybrid architectures. Our paper demonstrates that transformers can also been seen through the lens of dynamical systems, with a similar continuous extension as seen in neural ODEs.

Finally, the new methodology that we introduced is portable and widely applicable. Incidentally, it can also serve as a diagnostic method to highlight intrinsic differences between transformer layers. Fig. A8 to Fig. A10, for example, show significant deviations in the last layer (and to a lesser extent in the early ones). This suggests that these layers achieve a different kind of processing than intermediate layers, possibly following fine-tuning and/or re-alignment. It's not immediately obvious to us how these "anomalies" could be detected through a different approach.

## ACKNOWLEDGMENTS

This work was supported by the SciAI Center, and funded by the Office of Naval Research (ONR), under Grant Numbers N00014-23-1-2729 and N00014-23-1-2716.

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

# Appendix

## A  ADDITIONAL METHODS AND DERIVATIONS

### A.1  PSEUDO-SENTENCES

Random sample of 10-token pseudo-sentences (non-consecutive) extracted from *Walden*. Similar chunks, but of 50 tokens, were passed through GPT2 to form trajectories.

```
| not been made by my townsmen concerning my mode
| to pardon me if I undertake to answer some of
| writer, first or last, a simple and sincere
| would fain say something, not so much concerning
| Brahmins sitting exposed to four fires and looking
| more incredible and astonishing than the scenes which I daily
| and farming tools; for these are more easily acquired
|. How many a poor immortal soul have I met
| into the soil for compost. By a seeming fate
| as Raleigh rhymes it in his sonorous way
|il, are too clumsy and tremble too much
| the bloom on fruits, can be preserved only by
```

### A.2  LANGEVIN EQUATION DERIVATION

Starting from
$$\boldsymbol{x}(t+\tau) = \boldsymbol{R}(t+\tau)\boldsymbol{\Lambda}(t,\tau)\boldsymbol{R}(t)\boldsymbol{x}(t) + \boldsymbol{w}(t,\tau),$$
with $\boldsymbol{\Lambda}(t,\tau) = \boldsymbol{\Sigma}(t+\tau)\boldsymbol{\Sigma}^{-1}(t)$, and assuming now that $t,\tau$ are variables in $\mathbb{R}$, as $\tau$ goes to 0 we can approximate:

$$\boldsymbol{R}(t+\tau) \approx \boldsymbol{R}(t) + \tau\dot{\boldsymbol{R}}(t)$$

and

$$\boldsymbol{\Sigma}(t+\tau) \approx \boldsymbol{\Sigma}(t) + \tau\dot{\boldsymbol{\Sigma}}(t),$$

leading to:
$$\boldsymbol{\Lambda}(t,\tau) \approx \left(\boldsymbol{\Sigma}(t) + \tau\dot{\boldsymbol{\Sigma}}(t)\right)\boldsymbol{\Sigma}^{-1}(t) = \boldsymbol{I} + \tau\boldsymbol{\Sigma}^{-1}(t)\dot{\boldsymbol{\Sigma}}(t).$$

Hence:
$$\boldsymbol{R}(t+\tau)\boldsymbol{\Lambda}(t,\tau)\boldsymbol{R}(t)^{\top} \approx \left(\boldsymbol{R}(t) + \tau\dot{\boldsymbol{R}}(t)\right)\left(\boldsymbol{I} + \tau\dot{\boldsymbol{\Sigma}}(t)\boldsymbol{\Sigma}^{-1}(t)\right)\boldsymbol{R}(t)^{\top}$$
$$\approx \boldsymbol{I} + \tau\left(\dot{\boldsymbol{R}}(t)\boldsymbol{R}(t)^{\top} + \boldsymbol{R}(t)\dot{\boldsymbol{S}}(t)\boldsymbol{R}(t)^{\top}\right),$$

given that $\boldsymbol{R}\boldsymbol{R}^{\top} = \boldsymbol{I}$ and with $\boldsymbol{S}(t) = \mathrm{diag}\left(\ln\sigma_i(t)\right)$ and thus $\dot{\boldsymbol{S}}(t) = \mathrm{diag}(\dot{\sigma}_i/\sigma_i)$.

The variance of the noise term is given by:
$$\mathrm{var} = \alpha\exp(\lambda(t+\tau)) \approx \alpha\exp(\lambda t)(1+\lambda\tau).$$

The increment of variance over time $\tau$ is:
$$\delta[\mathrm{var}] = \alpha\lambda\exp(\lambda t)\tau.$$

This means the noise term can be expressed as:
$$\boldsymbol{w}(t,\tau) = \sqrt{\alpha\lambda\exp(\lambda t)\tau} \cdot \vec{\eta},$$

where $\vec{\eta}$ is a vector of standard Gaussian random variables.

Putting everything together:
$$\boldsymbol{x}(t+\tau) - \boldsymbol{x}(t) = \tau\left(\dot{\boldsymbol{R}}(t)\boldsymbol{R}(t)^{\top} + \boldsymbol{R}(t)\dot{\boldsymbol{S}}(t)\boldsymbol{R}(t)^{\top}\right)\boldsymbol{x}(t) + \sqrt{\alpha\lambda\exp(\lambda t)\tau}\,\eta(t).$$

And finally:

$$d\boldsymbol{x}(t) = \left( \dot{\boldsymbol{R}}(t)\boldsymbol{R}(t)^\top + \boldsymbol{R}(t)\dot{\boldsymbol{S}}(t)\boldsymbol{R}(t)^\top \right) \boldsymbol{x}(t)dt + \sqrt{\alpha\lambda \exp(\lambda t)}\, d\boldsymbol{w}(t),$$

with $d\boldsymbol{w}(t)$ a Wiener process.

### A.3   NUMERICAL INTEGRATION

Numerical integration of Eq. (3) requires to interpolate the singular vectors and values, and their derivatives, at non-integer times.

Interpolation of (scalar) singular values is straightforward. We use a polynomial interpolation scheme for each value, and compute the corresponding polynomial derivative. This yields $\dot{\sigma}_i(t)/\sigma_i(t)$ for every coordinate $i$ at any time $t \in [1, 24]$, and hence $\dot{\boldsymbol{S}}(t)$.

Interpolating sets of orthogonal vectors presents significant challenges. A rigorous approach involves performing the interpolation within the compact Stiefel manifold, followed by a reprojection onto the horizontal space Praveen et al. (2023). However, this method is computationally expensive and can introduce discontinuities, which are problematic for numerical integration. To address these issues, we used an approximation based on the matrix logarithm, which simplifies the process while maintaining an acceptable level of accuracy. To interpolate between $\boldsymbol{U}_1$ and $\boldsymbol{U}_2$ at $t_1, t_2$, we compute the relative rotation matrix $\boldsymbol{R} = \boldsymbol{U}_1^\top \boldsymbol{U}_2$ and interpolate using

$$\boldsymbol{U}(t) = \boldsymbol{U}_1 \exp_{\mathrm{M}}(\alpha \ln_{\mathrm{M}} \boldsymbol{R}). \tag{5}$$

where $\alpha = (t - t_1)/(t_2 - t_1)$ and with $\ln_{\mathrm{M}}, \exp_{\mathrm{M}}$ denoting the matrix logarithm and exponential, respectively.[17] This also yields the derivative $\dot{\boldsymbol{U}}(t) = [\boldsymbol{U} \ln_M \boldsymbol{R}]/(t_2 - t_1)$. Indeed:

$$\dot{\boldsymbol{U}} = \boldsymbol{U}_1 \cdot \frac{d}{dt} \exp\left( \alpha(t) \ln \boldsymbol{R} \right) = \boldsymbol{U}_1 \dot{\alpha} \ln \boldsymbol{R} \exp \alpha(t) \ln \boldsymbol{R} = \dot{\alpha} \boldsymbol{U} \ln \boldsymbol{R}.$$

## B   SUPPLEMENTARY FIGURES AND SCHEMATICS

### B.1   TRAJECTORY CLUSTERING

In Fig. A1, we show evidence of trajectory clustering in the latent space. In particular, all pilot tokens get transported away from the origin (or their starting point) by a comparable amount, resulting in narrow distributions along the first singular direction. Another signature of clustering is the fact that token positions at different times form distinct clusters, as showed by low-dimensional t-SNE representation (van der Maaten & Hinton, 2008).

### B.2   TRAJECTORY EXTRAPOLATION

In Fig. A2, we provide a schematic to explain the reasoning behind Eq. (1). *If the cluster rotated and stretched like a solid, the position of a point $\boldsymbol{x}'$ at $t'$ could be inferred exactly from it position $\boldsymbol{x}$ at $t$, using the formula outlined. However, unsurprisingly, the token ensemble does not maintain its topology and the points move around the clusters, requiring the stochastic term $\mathbf{w}$ injected in Eq. (1).

### B.3   SEPARABILITY OF TRUE AND EXTRAPOLATED POSITIONS

To characterize the similarity between the ensemble of true positions at $t + \tau$, $\boldsymbol{x}(t + \tau)$, from the positions extrapolated from $t$, $\tilde{\boldsymbol{x}}(t, \tau) = \boldsymbol{R}(t + \tau)\boldsymbol{\Lambda}(t, \tau)\boldsymbol{R}(t)^\top \boldsymbol{x}(t)$, we evaluate how much the two sets can be separated with a linear classifier. We train a Support Vector Machine Model with a linear kernel for each set of extrapolations $\{\tilde{\boldsymbol{x}}(t, \tau)\}$ (70/30 train/test). We then apply the classifier to predict whether points in the test set are true or extrapolated. In Table 1, we report results for the panels corresponding to Fig. 3. The accuracy of the classifier lies in the 50%-60% range, barely above random guessing (50%).

---

[17]$\exp_{\mathrm{M}}(\boldsymbol{A}) = \sum \boldsymbol{A}^k/k!$ and $\ln_M$ is the inverse function: $\ln_{\mathrm{M}} [\exp_{\mathrm{M}}(\boldsymbol{A})] = \boldsymbol{I}$.

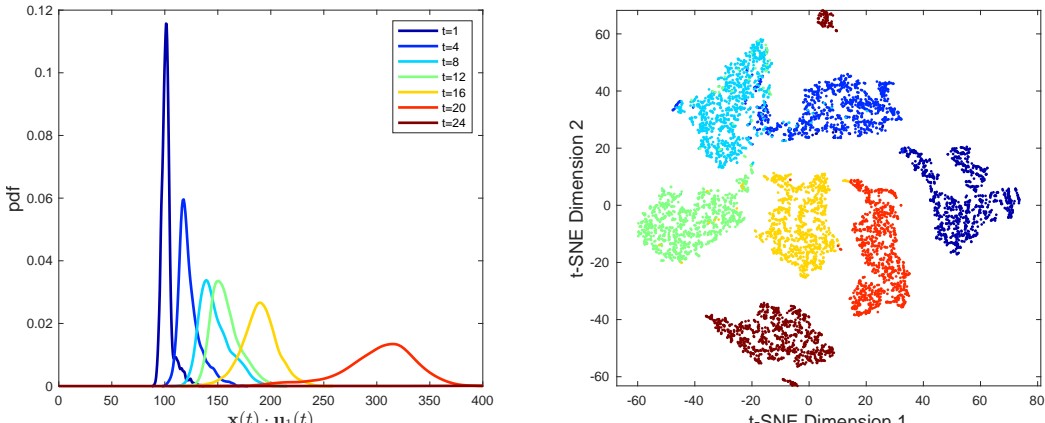

Figure A1: (Left) Distributions along the first singular vector at different times. (Right) Low-dimensional (t-SNE) visualization of the clustering of tokens, notably across different times. Same color legend.

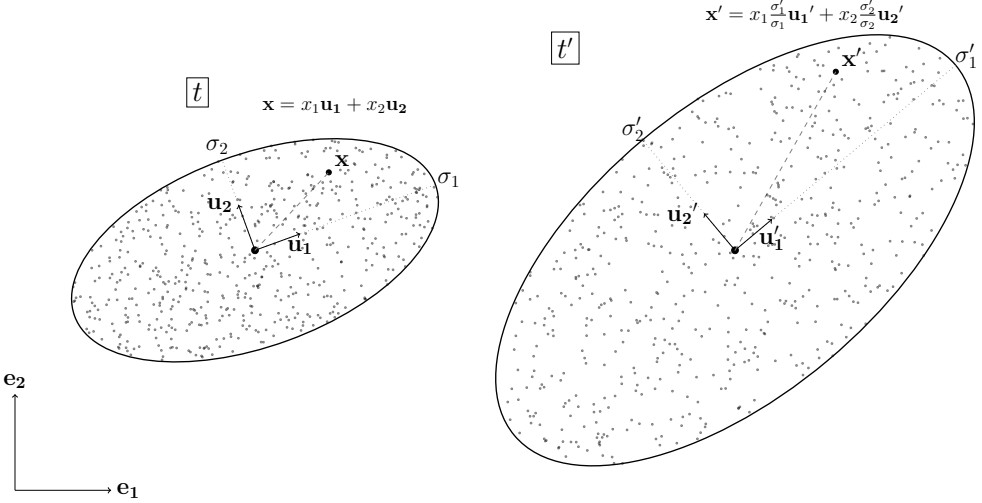

Figure A2: Extrapolation between $t$ and $t'$. The extrapolated location $\boldsymbol{x}'$ corresponds to the rotated and stretched position of $\boldsymbol{x}$. Given that $\vec{u} = \boldsymbol{R}\vec{e}$, $\vec{u}' = \boldsymbol{R}'\vec{e}$ and $\boldsymbol{R}^{-1} = \boldsymbol{R}^\top$, we have $\vec{e} = \boldsymbol{R}^\top \vec{u} = \boldsymbol{R}'^\top \vec{u}'$ and thus $\vec{u}' = \boldsymbol{R}'\boldsymbol{R}^\top \vec{u}$.

Table 1: Accuracy of linear classifier to separate $\boldsymbol{x}(t + \tau)$ and $\tilde{\boldsymbol{x}}(t, \tau)$, in percent.

|                  | $t = 12$ | $t = 14$ | $t = 16$ | $t = 18$ |
|------------------|----------|----------|----------|----------|
| $t + \tau = 13$  | 48       |          |          |          |
| $t + \tau = 15$  | 51       | 49       |          |          |
| $t + \tau = 17$  | 56       | 53       | 47       |          |
| $t + \tau = 19$  | 54       | 56       | 55       | 46       |
| $t + \tau = 21$  | 56       | 60       | 61       | 55       |

### B.4 Noise statistics

Fig. A3 provides additional details pertaining to the distribution of residuals $\delta_x$. Since they are many dimensions and time points, it gives only representative snapshots. It intends to substantiate the results that:

- the $\delta \boldsymbol{x}$ are Gaussian (Fig. A3A);
- the variance is exponential in $(t + \tau)$, with no dependency on $t$ (Fig. A3B);
- all components $\delta x_i$ of $\delta \boldsymbol{x}$ have the same distribution (Fig. A3C), i.e., isotropy;
- there are no spatial cross-correlations, i.e. $\langle \delta x_i \delta x_j \rangle = \delta_{ij}$ (Dirac function) (Fig. A3D).

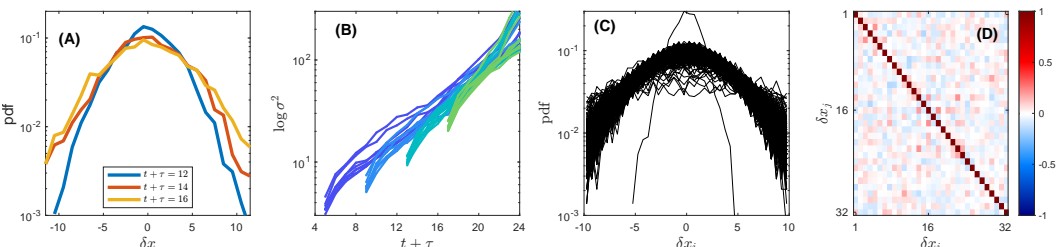

Figure A3: Statistics of $\delta \mathbf{x}$. (A) Empirical PDF of $\delta x_{42}(10, t + \tau)$, with $t + \tau = 12, 14, 16$. The curves appear Gaussian. (B) Variance of $\delta x_i$ for $i = 1 \ldots 8$, for $t = 4, 8, 12, 16$ and $t + \tau > t$. (C) Empirical PDF of $\delta x_i(12, 14)$ for $i = 1 \ldots 1024$. The curves are similar for almost all coordinates. (D) Cross correlations of $\delta x_i$ and $\delta x_j$.

### B.5 Details on noise aggregated statistics (Fig. 4)

Fig. A4 explains how the noise plots such as Fig. 4 are created. We use ensemble averages $\langle \ldots \rangle$ of the *absolute values* for $|\mu_i|, |\kappa_i|$ since we are interested in the average *distances* from 0.

### B.6 Null testing

Fig. A5 shows the trajectories of language vs gibberish, as well as the linear separability of the two ensemble. It also shows trajectories for an untrained GPT-2 shell, and a model with only the last 12 layers reinitialized.

### B.7 Results with other models

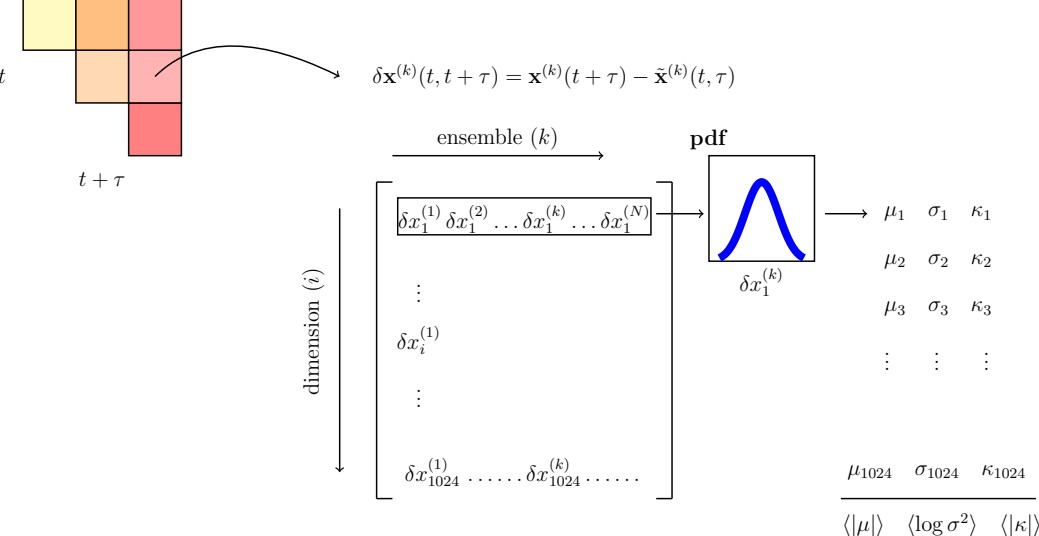

Figure A4: Schematic to explain the noise figures such as Fig. 4. Each square represents a summary statistics. Specifically, the square at $(t, t + \tau)$ represents the distribution of $\{\delta \boldsymbol{x}^{(k)}(t, t + \tau)\}_k$, with $k$ indexing individual tokens. The $\delta \boldsymbol{x}$ along each coordinate $i$ form a distribution, from which one can extract the corresponding $\mu_i, \sigma_i, \kappa_i$ (mean, variance, kurtosis). These 1D moments are then averaged along all coordinates $i$ ($\langle \mu_i \rangle_i$, etc.), forming the value displayed in the square.

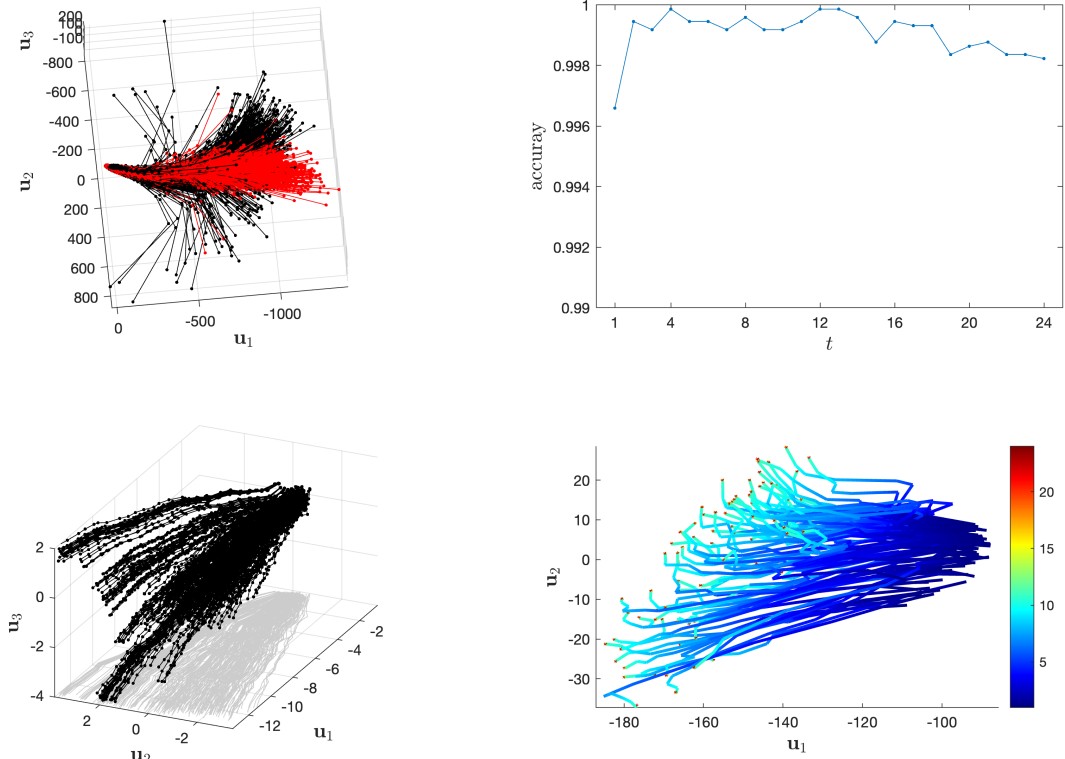

Figure A5: (Top-left) Trajectories of non-language (red) vs language (black), plotted in the same axes (10-token pseudo-sentences). (Top-right) Accuracy of linear separability between language and non-language for each layer. Obtained by training a Perceptron (train/test: 0.7/0.3; 14000 trajectories). (Bottom-left) Trajectories in the untrained GPT-2 model. They are transported in straight lines. (Bottom-right) Trajectories in the mixed model. After being transported by trained layers 1-12, the trajectories stop. Layers 13-24 with random weights do not transport tokens any further.

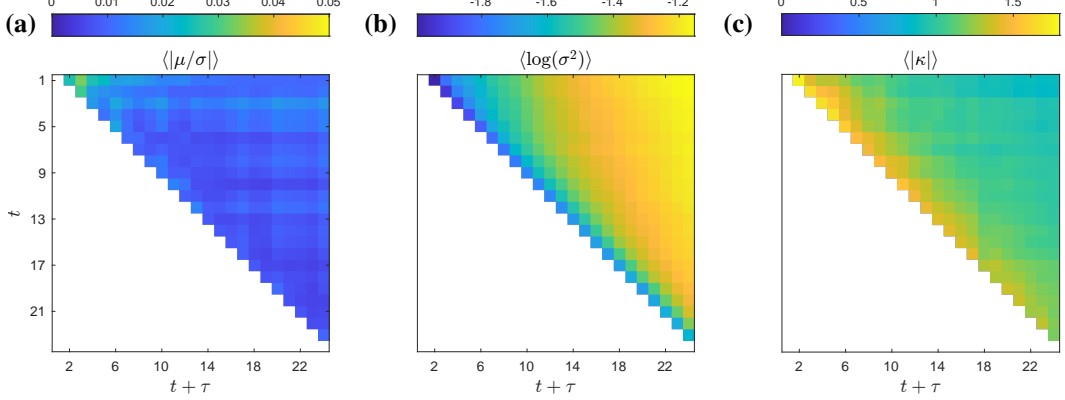

Figure A6: GPT-2 UNTRAINED. The averaged excess kurtoses $\langle|\kappa|\rangle$ fall in the 1–1.5 range, indicating strong non-gaussianity. The variance does not scale solely with $t + \tau$.

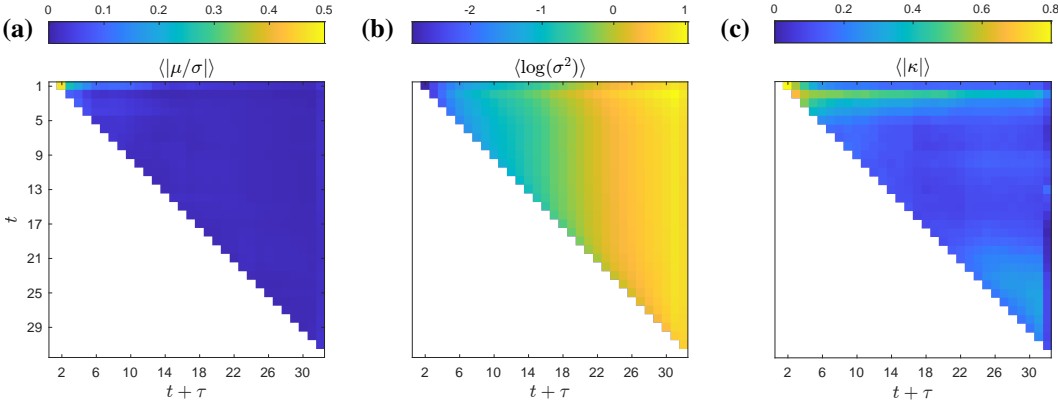

Figure A7: LLAMA 2 7B: noise statistics, $\delta\boldsymbol{x}(t, t+\tau) = \boldsymbol{x}(t+\tau) - \tilde{\boldsymbol{x}}(t, \tau)$, averaged $\langle \cdots \rangle$ over all Cartesian dimensions, for 1000 trajectories (50-token chunks). **(a)** Mean over standard deviation. **(b)** Logarithm of variance. **(c)** Excess kurtosis (0 means Gaussian).

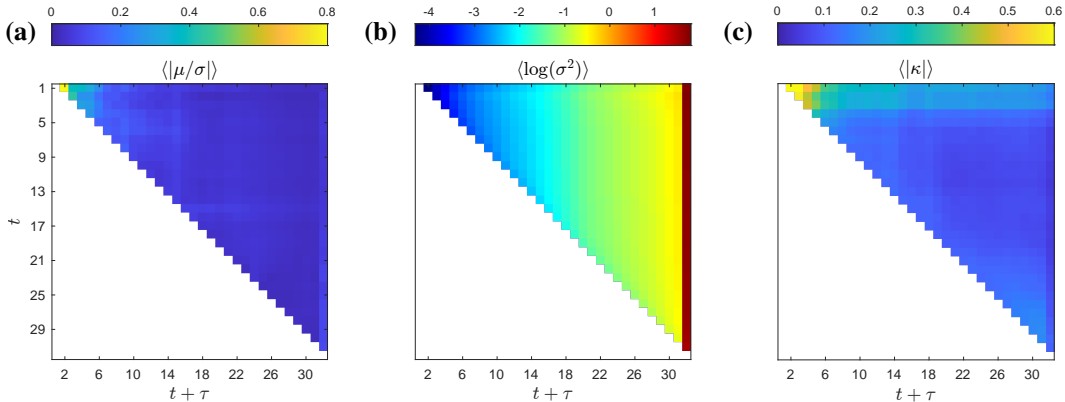

Figure A8: MISTRAL 7B V0.1. The last layer (32) appears to have an anomalously large variance.

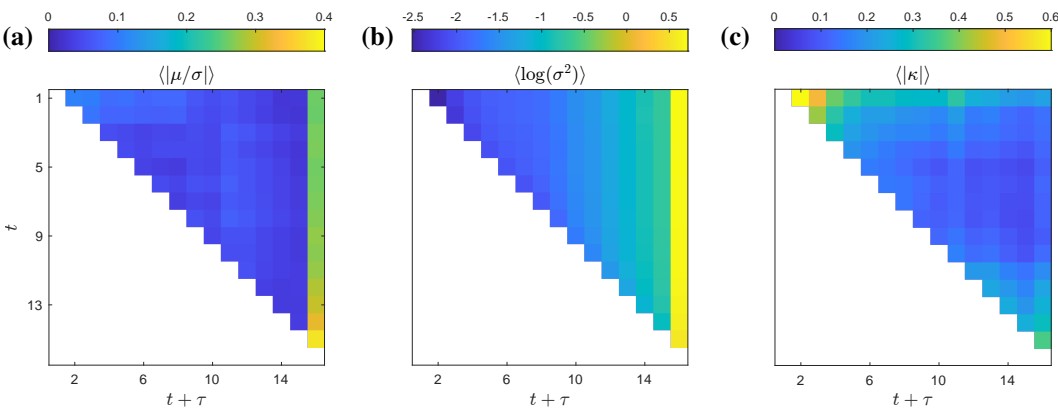

Figure A9: LLAMA 3.2 1B. This small model all present an out-of-distribution last layer.

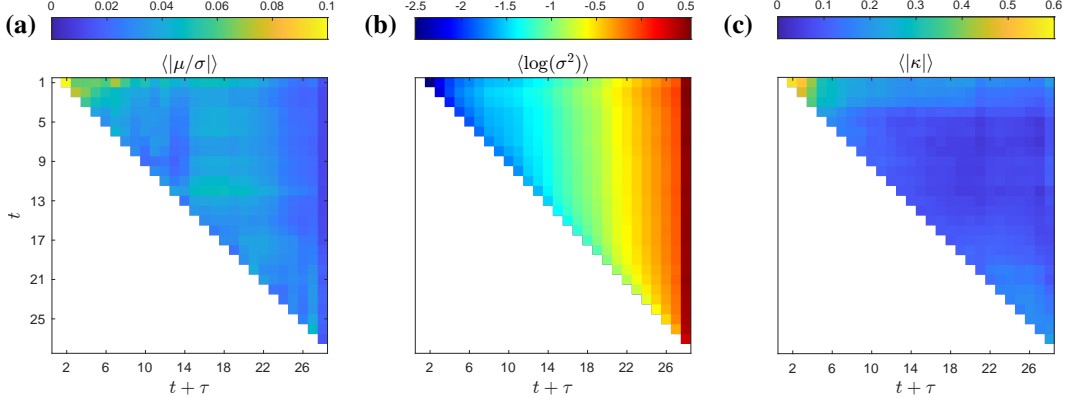

Figure A10:  LLAMA 3.2 3B. The last layer anomaly is also present.

