# OpenReview forum: "Lines of Thought in Large Language Models"
_ICLR.cc/2025/Conference — ICLR 2025 Poster_

### Official Review · Reviewer_UUNh · 2024-10-31

**Soundness:** 3
**Presentation:** 4
**Contribution:** 3
**Rating:** 8
**Confidence:** 3

**Summary:**

The authors describe a framework that allows tracing the path taken by the activations in latent space during a forward pass on some input text. They note that despite the complexity of computing a forward pass, the trajectory taken is rather simple, and the activations live on a low-dimensional manifold that moves through latent space as a function of the number of layers processed. They provide a stochastic model that describes the trajectory, with vastly fewer parameters than the network itself. The approximation closely agrees with the actual activations, and can be used to extrapolate the behavior of the model.

**Strengths:**

* Strong evidence provided to back up the claims made. The authors do a great job of explaining what it is they are measuring and why.
* The result is fascinating, computationally cheap to obtain, model agnostic, and is empirically backed by measuring the KL-divergence of the distribution after down-projecting the dimensionality.
* The theoretical justification on the proposed model is well motivated and explained in the appendix.
* See questions

**Weaknesses:**

* Some claims on the similarity of extrapolated token positions are a bit weaker, and are primarily based on visually comparing the extrapolated and actual activations projected down to a low-dimensional space. I would have liked to have seen a more concrete metric of comparison, together with some baseline values to compare against to indicate that they really are close.
* I would have liked to have seen some applications of this discovery, or at least conjectures for what it could be used for. For example, could one project down the weight matrices into the space provided, and obtain a network that has similar performance and is much smaller, at the expense of off-distribution task degradation? Essentially, a way to distil a pre-trained network down if we only care about performance on one task?
* I would elaborate on the potential use-case for a continuous interpolation between layers. Is there any sort of intuitive meaning we can assign to this?
* See questions

**Questions:**

Questions: (annotated with line number/section)
125: The vector for every token is projected to form the logits, not just the last vector. For inference, we only care about the logits in the last token'th position, but that doesn't mean the logits must only come from there.
130: Algorithm 1 is overkill. Just saying "we cache the activations between each transformer block in the sequence position of the last token over a forward pass for each psuedo-sentence $s_i$" would be sufficient. The algorithm is not doing anything interesting that the transformer would not already do during a forward pass (other than the caching of activations).
145: It is advised for ICLR to notate the last token position for layer $t+1$ as $\mathbf{E}^{t+1}_{:, end}$ rather than $\mathbf{E}(t+1)[:, end]$ (check the ICLR formatting guide).
152: The latent space is obviously spanned by a Cartesian basis. Any such space is.
155: You then revert to the ICLR standard notation for indexing a matrix that you did not use on line 145. Pick one and be consistent.
175-176: Replace "Because the Cartesian axes, $\mathbb{e}_i$, are unlikely to align with trajectories meaningful directions," with "There is no reason a priori we would expect the Cartesian axes to align to meaningful directions for the trajectories of the activations"
177: Concatenating the $\mathbb{x}_k(t)$? Over what dimension? $k$ or $t$ or something else?
154: Does it makes sense to plot the trajectories such that at each layer we use a different basis? Seems like you'd want to use the same basis throughout the entire forward pass if possible, as then each step along the trajectory is measured in terms of something different. I guess maybe the size of the principal component, with respect to the most important eigenvector is still interesting. This was partially addressed by Fig. 2a. showing that all the dimensions are important at some point during the forward pass, and that the low-dimensional manifold "wanders around" the entire latent space.


Section 3.2: At first glance this seems a reasonable choice of metric: The KL divergence between the normal output distribution, and the output distribution after having performed dimensionality reduction via SVD. Is this a metric that is defined elsewhere in the literature? Cite if so. I wonder if SVD is the best thing to do here: Is there a better choice of low-dimensional manifold that can only be obtained via a non-linear transformation? Perhaps if an autoencoder was trained that compresses/uncompresses the latent space on forward passes, could you squeeze the activations into an even lower dimensional space while preserving the KL divergence?
However, what isn't clear to me is that how much displacement of KL divergence is a lot, and why $K_0 = 256$ was the value for which "most" of the true distribution is recovered. What is "most"? What makes a KL-divergence of ~0.7 "not much"? It would be good to have some sort of sense of scale what the downstream effects are (e.g. measure the performance on some benchmark before and after throwing away 75% of the activations, and see how much "dumber" the model gets.)

Figure 2(c): What precisely is the baseline defined as? The KL divergence between the output distributions conditioned on two unrelated inputs, or with random noise injected into the unembedding matrix, or something else?

Figure 3: It's not clear how to compare the true and extrapolated positions. If the cluster comparison is desired, maybe colour one red, and the other blue, but both have transparency? So the purple region is the overlap? It's hard to see with the grey just covering the blue. I also would have liked a more principled metric to measure how well the model suggested in Equation (1) captures the dynamics of the activations, and what component remains unexplainable.

255-256: The rotation matrix $\mathbf{R}(t)$ was substituted out for $\mathbf{U}(t)$, but $\mathbb{\Lambda}(t,\tau)$ was not substituted out for $\mathbb{\Sigma}(t+\tau)\mathbb{\Sigma}^{-1}(t)$. I think either substitute both (my preference) or neither.

619: Forgot to bold $\Lambda$

---

> ### Author Response · Authors · 2024-11-17
>
> Thank you very much for the thorough review and very insightful comments. We are glad that you find our paper well-explained and our results fascinating!
> We have revised our manuscript to incorporate your suggestions and add appropriate clarifications. We respond to the questions below, and to the weaknesses shortly after, in the next comment.
>
> **Regarding the questions:**
>
> - 125: that's correct, all final embeddings can form logits, but only the last (right-most) embedding generates predictions for the next-token (intermediate logit for token n in teh sequence generates prediction for token n+1, which is already in the prompt). We have clarified that in the revisions.
> - 130: we agree that Algorithm 1 doesn't do anything remarkable; it is presented this way for clarity. Since the literature on LLMs is of interest for a broad community, we want to make sure the experimental basis to our finfings is conveyed clearly.
> - 145: good suggestion, we applied the ICLR guidelines.
> - 152: indeed; what we meant was defining the basis as E. We clarified the sentence.
> - 155: see above (145)
> - 175: good suggestion, we edited
> - 177: we have rephrased the whole paragraph, and details on how the singular vectors are obtained are presented in Section 2, "latent space bases".
> - 154: to be clear, the trajectories in Fig. 1 are plotted in a constant basis (the singular basis at t=24). But considering the bases at all times is useful to study the rotation and stretch of the ensemble (Fig. 2ab in particular).
>
> - Section 3.2:
>
> a) we are not aware of other instances where KL divergence between output distributions was used as a proxy for distance or dimensionality.
>
> b) SVD is a natural and simple transformation to investigate dimensionality. Using more complex, non-linear transformation is a clever suggestion which we keep in mind for upcoming projects; however, an auto-encoder might make things less interpretable?
>
> c) "How much is most?" Excellent question, and we should have been more rigorous in our writing. The problem is that KL divergence is difficult to interpret in practice. To avoid relying on absolute values (is 1.8 a lot for KL divergence?), we used relative values, first by comparing with the baseline of unrelated distributions (red lines), and by observing how much the average drops as a function of K. At K=256, $D_{KL}$ has dropped about 90% (from about 5 at K=1 to 0.5 at K=256), which is why we mentioned this (arbitrary) threshold. We clarified these points in our revisions.
>
> d) And indeed, what practically does it mean to have $D_{KL}$ = 0.5? Here's an illustrative downstream effect. If we compare the topK = 5 most likely output token with K=1024 vs K=256, the intersection of the two has average 2.5, meaning that half of the 5 most likely tokens at K=256 correspond to the most likely tokens at K=1024. While this is a concrete information, it's not quite as rigorous as a true probability metric, which is why we based our analysis on KL divergence between distribution (while now emphasizing the limitations of this approach).
>
> - Fig. 2: output distributions from unrelated inputs, as clarified in the revisions
> - Fig. 3: thank you for the suggestion, we are looking into a better visualization
> - 255: good point, we kept R(t) for conciseness and consistency with Eq. (1)
> - 619: corrected

---

> > ### Author Response · Authors · 2024-11-20
> > **[continuing the previous comment]**
> >
> > **Regarding the weaknesses:**
> >
> >
> > - **Metric for comparing extrapolated token positions to true ones:**
> > This is a very good remark, which we should have discussed more carefully earlier. Rigorous comparison of point distributions in high-dimensional spaces is a challenging exercise. Comparing marginal distributions, which is essentially what Fig. 3 does, is possible but often insufficient. We had also considered using Maximum Mean Discrepancy, but properly interpreting output values for significance is convoluted.
> > Upon reflection, we propose to use a **classifier** to evaluate whether the extrapolated and true ensembles can be linearly separated. If the classifier's accuracy (upon training on a subsample) is only slightly above 50%, it means the two sets are very similar.
> > By using a Support Vector Machine model with a linear kernel, we find an accuracy between 50%-51% when $\tau = 1$ (comparing across a single layer). Unsurprisingly, the accuracy increases with \tau as the extrapolative power of the model in Eq. (1) decreases with the number of intermediate layers. For $\tau = 10$, accuracy is about 60%. We emphasize that this is a comparison at the ensemble level, and point-wise discrepancy is described by the statistics reported in Fig. 4.
> > We are adding these details to the revisions.
> >
> >
> >
> > - **Applications of the discovery:**
> > Very good point; what could be the applications of our findings? Our dynamical system approach reveals a surprising dimensionality reduction of token embeddings. It suggests, notably, that the true "meaning" of token is contained within individual variability (possibly orthogonal to the average pathway commonly followed by all LoTs).
> > Our model in Eq. (3-4) is also merely a first-order approximation, which could be extended to more complete and precise equations, where the "noise term" becomes smaller and smaller. Eventually, we anticipate the possibility for hydrid architectures where the deterministic part of trajectories is delegated to a small system of equations, while the variable part, where most meaning is encoded, is handled by a neural network; potentially with many fewer weights.
> >
> >
> >
> > - **Potential use-case for the continuous interpolation between layers:**
> > also a great question. In the past, the neural Ordinary Differential Equation paradigm (Chen et al, 2018) showed that converting a discrete neural network into a continuous dynamical system had many advantages. Notably, it offers opportunities for compression and stability, while pointing towards efficient hybrid architectures.
> > Our paper demonstrates that transformers can also been seen through the lens of dynamical systems, and continuously extended. It would NOT have been the case if, for example, we had found that trajectories looked more diffusive, and with jagged paths, which would have been at odds with the proposed continuous description.

---

> ### Author Response · Authors · 2024-11-22
>
> As the author-reviewer discussion period is coming to an end (November 26th, this coming Tuesday), we would like to make sure the Reviewer doesn't have any further comments or questions. After that, we may not have the opportunity to continue this discussion.
>
> We thank the Reviewer for many valuable insights, which have helped up to substantially strengthen our paper.We hope that we have adequately responded to your questions.

---

> > ### Comment · Reviewer_UUNh · 2024-11-26
> >
> > We thank the authors for their response.
> > We feel our concerns were adequately addressed, and do not have any further questions or concerns.
> > Unless there is disagreement on your end, we will leave our score as-is.

---

> > > ### Author Response · Authors · 2024-11-26
> > >
> > > We are pleased to read we were able to address your questions and concerns adequately. Thank you again for your comments.

---

### Official Review · Reviewer_k1Ts · 2024-11-02

**Soundness:** 3
**Presentation:** 3
**Contribution:** 3
**Rating:** 6
**Confidence:** 3

**Summary:**

This paper studies the trajectory of token embedding across layers. Inspired by dynamic system, they model the trajectories as diffusive process with a linear drift and a modified stochastic component.

**Strengths:**

1.The idea of this paper is kind of interesting.

**Weaknesses:**

1. the Gaussian assumption seems to not hold in early layers.
2. It is unclear if the same type of paths would hold for larger and more complex model as we already see problems with newer model like LLaMA-3. I think it makes sense to model the intermediate layers with diffusion process but early and last layers might not work not well.
3. The theory here does not lead to any practical predictions. For example, can you use this model to predict next token?

**Questions:**

1. It is unclear to me what does this sentence (”the uncertainty (or stochasticity) introduced here accounts only for the loss of information of considering the token without its prompt.”) in footnote 8 mean?
2. For figure 3, why u_2 and u_3 without u_1?

---

> ### Author Response · Authors · 2024-11-17
>
> Thank you very much for the comments. We respond to the specific points below, and revised the manuscript accordingly.
>
> **Regarding the questions:**
>
> 1) Regarding footnote 8, we revised the sentence for clarity: "We emphasize that prompt trajectories are completely deterministic; the stochastic component introduced in the model accounts for the fact that we perform a linear extrapolation based only on a token's position at a certain time, which unsurprisingly deviates from the true position obtained from processing the full prompt with transformer layers."
>
> 2) Regarding the projection planes in figure 3: good observation! We are showing the (u2, u3) plane simply for clarity, as the point clouds are more stretched along u1, making the plots less readable.
>
>
> **Regarding weaknesses:**
>
> 1) Regarding the Gaussian assumption in early layers: That's correct! As we point out in the paper, our approach reveals significant deviations especially in the last layers, but also in the early layers (to a lesser extent), as seen in Fig. A8 to A10. It seems like an interesting finding that early and final layers are "different", and might suggest that these layers achieve a different kind of processing than intermediate layers, possibly following fine-tuning and/or re-alignment. It's not immediately obvious to us how these "anomalies" could be detected through a different approach. Therefore, we even envision that our approach could serve as a diagnostic method to highlight intrinsic differences between transformer layers.
>
> 2) About whether our findings would hold for larger and more complex models: it's an interesting point. Here we focus on models in the ~1-10B parameters for insight. For larger open-source models, typically 70B or 400B, the increased number of dimensions and layers require much larger input ensembles, beyond the scope of this study. However, larger open-source models have the same architecture as smaller ones. Besides, the Llama 3.2 1B and 3B where the newest open-source models at time of writing.
>
> 3) Can we use this model for next token prediction?
> This is a very fair question.
> Our model in Eq. (3-4) is merely a first-order approximation, which could be extended to more precise equations where the "noise term" becomes smaller and smaller. We envision this could lead to the possibility for hydrid architectures where the deterministic part of trajectories is delegated to a small system of equations, while the variable part, where most meaning is encoded, is handled by a specialized neural network; potentially with many fewer weights.
> In addition, our results demonstrate that transformers can also been seen through the lens of dynamical systems, and continuously extended. (It would NOT have been the case if, for example, we had found that trajectories looked more diffusive, with a jagged path, which would have been at odds with the proposed continous description.) In the past, the neural ODE paper (Chen et al, 2018) showed that converting a discrete neural network into a continuous dynamical system had many advantages. Notably, it offers opportunities for compression and stability, while pointing towards efficient hydrib architectures.
>
> Finally, we would like to emphasize that this is a work on interpretability, and our primary objective is to better understand emergent properties of LLMs (as alluded to in the introduction). Since a language model is basically a mathematical object which maps a (vectorized) prompt onto a discrete probability distribution (on the vocabulary), we endeavor to offer insight into the kinds of mathematical transformations that LLMs perform within the latent space.

---

> ### Author Response · Authors · 2024-11-22
>
> As the author-reviewer discussion period is coming to an end (November 26th, this coming Tuesday), we would like to make sure the Reviewer doesn't have any further comments or questions. After that, we may not have the opportunity to respond to the comments.
>
> We hope that we have adequately responded to your questions.
> Again, we thank the Reviewer for valuable insights, which have helped up improve our paper, notably regarding its motivation and implications.

---

> > ### Comment · Reviewer_k1Ts · 2024-11-27
> >
> > Thanks for your responses! I still feel like my original score is a faire assessment of the work which is still leaning acceptance and I overall think the paper is a useful contributions to the community. Having said that, while I agree that the work is focused on interpretability, this paper is still different from traditional interpretability research. Mainly, I don't know if such interpretability is useful. In other words, the line of thoughts is not tied to any semantic structure or meanings.

---

> > > ### Author Response · Authors · 2024-11-27
> > > **Response to Reviewer k1Ts**
> > >
> > > Thank you for your reply and additional comments.
> > >
> > > We absolutely agree that we are proposing a new framework for interpretability, although not unrelated to other recent works (see, for example, [1] and [2] below). Our approach focuses on describing the mathematical transformations performed by LLMs to simplify the vast complexity of billions of weights and biases.
> > >
> > > While traditional interpretability research (which we succinctly review in our introduction) has begun to elucidate aspects of feature representation, identified task-specific circuits and induction heads, and uncovered signatures of "world models," it generally focuses on specific properties. For example: Is a specific kind of information (true/false, city location [3], etc.) encoded in embeddings, and how? Which neurons correspond to which concepts?
> > >
> > > In contrast, our lines-of-thought/dynamical system approach is agnostic to semantics and syntax, making it more general and widely applicable. This is particularly valuable because LLMs often exhibit unexpected emergent capabilities, notably some that go beyond language generation (e.g., time series prediction, in-context learning, etc.).
> > >
> > > We believe that by describing the path taken by an embedding toward its mapping into a next-token prediction vector, we are paving the way for a more systematic analysis of the transformations that data undergo within the network.
> > >
> > >
> > >
> > > [1] Geshkovski et al., "The Emergence of Clusters in Self-Attention Dynamics." In Advances in Neural Information Processing Systems, volume 36, 2024.
> > >
> > > [2] Lu et al., "Understanding and Improving Transformers from a Multi-Particle Dynamic System Point of View." arXiv:1906.02762.
> > >
> > > [3] Wes Gurnee and Max Tegmark, "Language Models Represent Space and Time." arXiv:2310.02207, 2023.

---

### Official Review · Reviewer_u8bS · 2024-11-04

**Soundness:** 2
**Presentation:** 2
**Contribution:** 2
**Rating:** 6
**Confidence:** 3

**Summary:**

This work studies the dynamics of the embedding space as the input tokens go through the layers of a transformer architecture. The authors break the input prompt into sequences of tokens and study the embeddings of the last token as it propagates through the layers. Using PCA-like projections, it's shown that these embeddings approximately lie in a low-dimensional manifold. Then, by rephrasing these trajectories using linear approximations, the authors attempt to model it via Langevin dynamics which gives rise to the standard Fokker-Planck probability flow. One of the main conclusions is that transformers can be "distilled" into few parameters. To validate their assumptions, the authors experimentally study these trajectory on non-language inputs and untrained models and show that their ideas weakly hold. The target audience are people interested in the physics and interpretability of large language models.

**Strengths:**

- There has been a flurry of works trying to understand the inner workings of LLMs. Therefore, this direction is relevant and interesting.

- This work studies this problem from a unique flow-based perspective by studying the dynamics of the embeddings as they evolve in the layers. The perspective is novel to the best of my knowledge and may potentially lead to a new perspective or algorithm to improve interpretability of LLMs.

**Weaknesses:**

- While potentially interesting, the paper feels too vague and very high-level without any concrete theoretical or experimental contributions.

- No new theoretical contributions are made, other than standard langevin dynamics formulations of their ideas. The projection to lower dimensions and linear approximations are also somewhat too lossy, as the authors note, so it's not clear how well the observations here actually hold in real life.

- Experiments seem limited to a few models and as the authors note, the last layer seems to form an outlier for the Mistral 7B and the Llama models. While the authors suggest some reasons for this, it's not inherently clear why these happen and bring the central hypotheses into question.

**Questions:**

Some questions were raised above. In addition

- This work reminds me of neural ODEs. Are there any connections between them and the viewpoint this paper explores?

---

> ### Author Response · Authors · 2024-11-17
>
> Thank you very much for the comments. As you kindly pointed out in the review, our work furnishes a means for gaining insight on some crucial mechanisms encoded in trained LLMs, and does this from the novel perspective of token trajectories. More concretely, we propose to approximate the mathematical transformation (from input prompt to next token prediction) realized by a stack of transformer layers by a simple set of equations.
>
>
> **Regarding the weaknesses:**
>
> 1) About the vagueness and lack of concrete contributions: We do believe that our paper provides significant new insights into how transformer layers operate on embedded tokens. We show that trained LLMs transport these embeddings across the latent space, not in any chaotic/unpredictable way, but rather along a well-defined manifold of low dimension; wherein the dynamics can be described with few parameters, despite it having emerged from millions/billions of weights. To our knowledge, this was not known until now. This observation also holds across a variety of language models, suggesting universality, which seems quite remarkable.
>
> 2) About how well observations hold in real life: Maybe there is a misunderstanding about what is meant by "in real life", but our experiments are based on open-source models that are widely used (as is) for chat and completion. Besides, the low dimensionality of trajectories is a substantial mathematical observation, with potential concrete implications. Indeed, our model in Eq. (3-4) is merely a first-order approximation, which could be extended to more complete and precise equations, where the "noise term" becomes smaller and smaller. We anticipate the possibility for hybrid architectures where the deterministic part of trajectories is delegated to a small system of equations, while the variable part, where most meaning is encoded, is handled by a neural network with many fewer weights.
>
> 3) About the limitation to few models: Our experiments investigate an extensive suite of models: GPT-2 medium, Mistral 7B, Llama 2 7B, Llama 3.2 1B and 3B (the last two being SoTA models released in September 2024). Besides, this is not a benchmarking paper, but rather an in-depth study of LLMs' essential and universal features.
> The last layer anomaly does not weaken the central hypothesis, as the low-dimensional dynamics holds for all other layers. Rather, our theory reveals that the last layer in fine-tuned/realigned models seem to exhibit a different pattern from all others, which might not be trivial to see otherwise. To use a simple illustration, it is analogous to performing a linear regression over time (ie, layers), and remarking that the points are aligned at all times except the last timepoints (and to a lesser extent the early ones too), for some models.
> In other words, our approach also provides a **diagnostic** to identify layers with anomalous features. This seems to be of interest, and definitely worthy of further investigation to detect these phenomena in pre-trained models.
>
>
> **Regarding the neural ODE similarity:**
> This is an excellent comparison and a very valuable comment, which we incorporate in our revisions. Briefly, Chen et al. (2019) trained a neural network to approximate trajectories based on the coarse-grained trajectory data generated by the original model. Here, we find an explicit theoretical model to do it. While it is less accurate, it provides a deeper insight into the inner workings of LLMs, as the neural ODE remains a "black box" (per the author words).
> Further, the neural ODE paradigm showed that converting a discrete neural network into a continuous dynamical system had many advantages. It offers opportunities for compression and stability, while pointing towards efficient hydrib architectures.
> Our paper demonstrates that transformers can also been seen through the lens of dynamical systems, and continuously extended. (It would NOT have been the case if, for example, we had found that trajectories looked more diffusive, with a jagged path, which would have been at odds with the proposed continous description.) This possibly paves the way for similar hybrid architectures for transformer-based networks.
>
>
> We have revised our manuscript to include the points made above, especially in the conclusion, and thank the reviewer for the valuable inputs.

---

> > ### Comment · Reviewer_u8bS · 2024-11-26
> > **Response to rebuttal**
> >
> > I thank the authors for their clarifications to me and the other reviewers. In light of this, I'm willing to increase my score.

---

> > > ### Author Response · Authors · 2024-11-26
> > >
> > > Thank you for taking the time to consider our responses. We are glad we were able to address your comments adequately.

---

> ### Author Response · Authors · 2024-11-22
>
> As the author-reviewer discussion period is coming to an end (November 26th, this coming Tuesday) we would be happy to continue this discussion with the Reviewer and answer potential additional questions. After that, we may not have the opportunity to respond to the comments.
> We hope that our comments addressed your concerns and clarified our contributions. Again, we thank the Reviewer for valuable insights, which have helped up improve our paper, notably regarding its motivation and implications.

---

### Official Review · Reviewer_AKd3 · 2024-11-05

**Soundness:** 3
**Presentation:** 2
**Contribution:** 2
**Rating:** 6
**Confidence:** 3

**Summary:**

The paper aims to study the statistical properties of what they call *lines of thought*; trajectories traced by the embedded tokens through the latent space while traversing successive transformer layers. The key observation is that independent trajectories cluster along a low-dimensional manifold, and that their paths can be approximated using a simple dynamics model.

**Strengths:**

- I find the question posed, and the consequent findings very interesting

- I really appreciate the development of a linear approximation to the distribution of trajectories.

**Weaknesses:**

- One of my biggest gripes with the paper is, while I wanted to be excited about the findings, a recurring question that I had was "why should I care"? It is my opinion that the authors should invest in a motivation for why the reader should care about the presented findings. It's not clear to me what the takeaways are, or more concretely, how we can utilize the observation to develop better LLMs for instance.

- A second gripe, which perhaps goes hand-in-hand with my first one, is that I often felt the paper could have used a bit more hand holding, or even been organized better. To make this concrete, in Section 3.1, the findings are presented first and then the methodology employed to find them was presented, which to me was a bit confusing. Section 3.2 starts abruptly with the sentence "The fast decay of the singular values...". Section 3.4 is titled "Langevin Dynamics..." with no mention of Langevin Dynamics. Furthermore, I often found myself having to reference many different figures in different parts of the paper while reading a single paragraph. All of this made it harder to follow the paper closely.

- (Stylistic nitpicking) I think some of the writing maybe embellishes on details that maybe are not very important e.g. the methodology section starts with what I believe to be details that can be abstracted away. I also think the footnotes are somewhat excessively used.

- I'm not really sure what is meant by "pilot" in this context. My understanding is that a pilot is  "done as an experiment or test before introducing something more widely."

- The authors claim that the pseudo-sentences are "independent" (line 171) which is not immediately clear to me given that the pseudosentences are chunks produced from a single piece of writing.

- I find the notation used in the algorithm confusing. Why do we sometimes use parentheses and other times square brackets? Also, wouldn't indexing with t+1=25 at  the last iteration be undefined?

**Questions:**

In addition to the concerns raised in the weaknesses,

- I find the point being made in the paragraph between lines 195-201 very interesting, but I am then immediately confused by the following paragraph. Could you please clarify this?

- Could you please clarify the paragraph between lines 248-251?

---

> ### Author Response · Authors · 2024-11-17
>
> Thank you for the detailed and thoughtful review. We appreciate your sentiment that several aspects of the paper are "very interesting", including its findings, but realize now that our presentation and motivation could be improved. We are revising the paper accordingly and now offer specific responses below.
>
> **Regarding the weaknesses:**
>
> 1) Regarding "why should I care?": This is indeed a very fair point, and something that we aim to better explain in the revised manuscript. We respond in-depth at the end of this response.
>
> 2) Regarding clarity/style:
> - We edited Paragraphs 3.1 and 3.2 for clarity, as suggested.
> - In paragraph 3.4, we clarified that Equation 3 is a Langevin equation.
> - Regarding the order of figures: several figures referenced are "Appendix Figures", which is why they appear later in the paper. We now refer to them as Fig. A1 - A10, hoping that it makes the flow of the paper easier to follow.
> - About the definition of 'pilot': we defined it in the Methods as the last embedding in the input prompt (line 125). It is the embedding that eventually generates the next-token distribution. We better emphasize this point within our revised manuscript.
> - About the independence of pseudo-sentences: we agree, but while pseudo-sentences may not be semantically independent, they are nonetheless non-overlapping. Thus, they don’t contain any (non-trivial) identical sub-sequence of tokens, which could a priori make their paths correlated. We clarify this point in the revisions.
> - Algorithm: Thanks for the suggestion, we have updated the paper to use the ICLR formatting guidelines to denote function arguments (superscripts) and array entries (subscripts)
>
>
> **Regarding the questions:**
>
> 1) Clarification of paragraph 202-207:
> it emphasizes that the low-dimensional manifold generated by trajectories is **curved**; in other words, the manifold cannot be contained in a subset of the Cartesian basis. A simple illustration of this notion would be to consider a sphere (i.e., hollow ball), which is intrinsically a 2D object that is embedded in a 3D space (imagine the point of view of an ant walking on the surface of this sphere); this is in contrast to a plane, which is **flat**; the hypothetical ant's intrisic coordinates would be consistent with those of the global embedding space.
>
> 2) Clarification of the paragraph between lines 248-251:
> it should have referred to Figure 7 (now Figure A2), which illustrates the deformation described and "solid-like" behavior. Thank you for pointing this out, we have corrected.
>
> **Back to "why should I care?"**
>
> Let’s preface what follows by highlighting that this is a work on interpretability, and our primary objective is to better understand emergent properties of LLMs (as alluded to in the introduction).
> Since a language model is basically a mathematical object which maps a (vectorized) prompt onto a discrete probability distribution (on the vocabulary), we endeavor to offer insight into the kinds of mathematical transformations that LLMs perform within the latent space. Besides introducing a new methodology that is portable and widely applicable, our contributions concern the low-dimensionality of embedding trajectories, and their possible extension to smooth continuous paths; this has some interesting implications, that follow, below.
>
> [to be continued in next comment]

---

> > ### Author Response · Authors · 2024-11-17
> > **[continuing the previous comment]**
> >
> > **Low dimensionality:**
> > For interpretability, finding low-dimensional structures is consequential. It is one of the most efficient ways to break down the inherent complexity of large models into more elementary constituents. Our dynamical system approach reveals a surprising dimensionality reduction of token embeddings. It suggests, notably, that the true "meaning" of token is contained within individual variability (possibly orthogonal to the average pathway commonly followed by all LoTs).
> > Our model in Eq. (3-4) is also merely a first-order approximation, which could be extended to more complete and precise equations, where the "noise term" becomes smaller and smaller. Eventually, we anticipate the possibility for hydrid architectures where the deterministic part of trajectories is delegated to a small system of equations, while the variable part, where most meaning is encoded, is handled by a neural network; potentially with many fewer weights.
> >
> > **Continuity:**
> > Our theoretical model (Eq. 3-4) not only reveals low-dimensionality, but also extends token trajectories to continuous paths. In the past, the neural ODE paper (Chen et al, 2018) showed that converting a discrete neural network into a continuous dynamical system had many advantages. Notably, it offers opportunities for compression and stability, while pointing towards efficient hydrib architectures.
> > Our paper demonstrates that transformers can also been seen through the lens of dynamical systems, and continuously extended. (It would NOT have been the case if, for example, we had found that trajectories looked more diffusive, with a jagged path, which would have been at odds with the proposed continous description.)
> >
> > **Diagnosis:**
> > Finally, we also remark that our approach, incidentally, can serve as a diagnostic method to highlight intrinsic differences between transformer layers. Fig. A8 to A10, for example, show a significant deviation from our model in the last layer (and to a lesser extent in the early ones). This suggests that these layers achieve a different kind of processing than intermediate layers, possibly following fine-tuning and/or re-alignment. It's not immediately obvious to us how these "anomalies" could be detected through a different approach.
> >
> > We have revised our manuscript to include the points made above and thank the reviewer for the comment that led us to include these additional explanations.

---

> ### Author Response · Authors · 2024-11-22
>
> As the author-reviewer discussion period is coming to an end (November 26th, this coming Tuesday) we would be happy to receive the Reviewer's responses, and answer any possible additional questions. After that, we may not have the opportunity to respond to the comments.
> We hope that our comments addressed your concerns and clarified our contributions. Again, we thank the Reviewer for valuable insights, which have helped up improve our paper, notably regarding its motivation and implications.

---

> > ### Comment · Reviewer_AKd3 · 2024-11-26
> >
> > Thank you for responding to my concerns, and apologies for my late reply. I have raised my score.

---

### Author Response · Authors · 2024-11-17

We have received detailed and insightful comments from all reviewers, and very much appreciate their feedback. Thank you!
We have responded to each review specifically, and made a number of substantial revisions to the manuscript (marked in orange in the PDF).

**It appears that all reviewers praised the novelty of our approach, and found our objectives and findings of great interest.**

Many questions were submitted, often asking for clarifications or further details, which we made sure to address and incorporate in our revisions.

**It seems like the main reservation across most reviews is about concrete implications of our results.** "Why should we care? Can we use the results to make better LLMs? Can we use the model to predict next token? Are there prospects for distillation?"

In retrospect, we realize we should have motivated our work much more robustly. While our focus is on interpretability, understanding emergent properties of LLMs, there are also several practical implications to our work.

We responded as follows, and revised our manuscript accordingly, especially in the conclusion.

**Methodology and contributions:**
Since a language model is basically a mathematical object which maps a (vectorized) prompt onto a discrete probability distribution (on the vocabulary), we endeavor to offer insight into the kinds of mathematical transformations that LLMs perform within the latent space. Besides introducing a new methodology that is portable and widely applicable, our contributions concern the low-dimensionality of embedding trajectories, and their possible extension to smooth continuous paths; this has some rather interesting implications, that follow, below.

**Low dimensionality:**
For interpretability, finding low-dimensional structures is consequential. It is one of the most efficient ways to break down the inherent complexity of large models into more elementary constituents. Our dynamical system approach reveals a surprising dimensionality reduction of token embeddings. It suggests, notably, that the true "meaning" of token is contained within individual variability (possibly orthogonal to the average pathway commonly followed by all LoTs).
This is also merely a first-order approximation, which could be extended to more complete and precise equations, where the "noise term" becomes smaller and smaller. Eventually, we anticipate the possibility for hybrid architectures where the deterministic part of trajectories is delegated to a small system of equations, while the variable part, where most meaning is encoded, is handled by a neural network; potentially with many fewer weights.

**Continuity:**
Our theoretical model (Eq. 3-4) not only reveals low-dimensionality, but also extends token trajectories to continuous paths. In the past, the neural ODE paradigm introduced by Chen et al. (2019) showed that converting a discrete neural network into a continuous dynamical system had many advantages. Notably, it offers opportunities for compression and stability, while pointing towards efficient hybrid architectures.
Our paper demonstrates that transformers can also been seen through the lens of dynamical systems, with a similar continuous extension as seen in neural ODEs. (It would NOT have been the case if, for example, we had found that trajectories looked more diffusive, with a jagged path, which would have been at odds with the proposed continous description.)

**Diagnosis:**
Finally, we also remark that our approach, incidentally, can serve as a diagnostic method to highlight intrinsic differences between transformer layers. Fig. A8 to A10, for example, show a significant deviation from our model in the last layer (and to a lesser extent in the early ones). This suggests that these layers achieve a different kind of processing than intermediate layers, possibly following fine-tuning and/or re-alignment. It's not immediately obvious to us how these "anomalies" could be detected through a different approach.

---

### Meta-Review · Area_Chair_vLyT · 2024-12-20

**Metareview:**

This paper describes an interesting phenomena in language models about the structure of the dynamics of residual stream updating across transformer layers. Reviewers found the idea interesting, the presentation clear, and were overall positive. However, all reviewers were unclear on the importance of this observation. It does not seem useful as either a subroutine in an engineering application, nor as a test of some scientific hypothesis. That is, no reviewer was clear on what it would mean to build on the results here. I strongly encourage the authors to elaborate on this aspect of the results in order to achieve significant impact.

**Additional Comments On Reviewer Discussion:**

See above

---

### Decision · Program_Chairs · 2025-01-22

Accept (Poster)